# Enzymatic Inhibitors of Aspartyl Protease EAP1 and Xylanase SRXL1 from *Sporisorium reilianum* Isolated from Corn Seeds

**DOI:** 10.3390/ijms26209974

**Published:** 2025-10-14

**Authors:** Yusiri Velázquez-Juárez, Alejandro Téllez-Jurado, Macaria Hernández-Chávez, Lourdes Villa-Tanaca, Martha Patricia Falcón-León, Yuridia Mercado-Flores

**Affiliations:** 1Laboratorio Nacional Conahcyt de Investigación y Servicios para la Productividad del Campo, Universidad Politécnica de Pachuca, Carretera Pachuca-Cd. Sahagún km 20, Ex-Hacienda de Santa Bárbara, Zempoala 43830, Mexico; yusivelazquezj@micorreo.upp.edu.mx (Y.V.-J.); alito@upp.edu.mx (A.T.-J.); 2Laboratorio de Optomecatrónica y Energías, Unidad Profesional Interdisciplinaria de Ingeniería Campus Hidalgo (UPIIH), Instituto Politécnico Nacional, Distrito de Educación, Salud, Ciencia, Tecnología e Innovación, San Agustín Tlaxiaca 42162, Mexico; mhernandezch@ipn.mx; 3Laboratorio de Biología Molecular de Bacterias y Levaduras, Departamento de Microbiología, Escuela Nacional de Ciencias Biológicas, Instituto Politécnico Nacional, Prol. de Carpio y Plan de Ayala, Col. Sto. Tomás, Ciudad de México 11340, Mexico; mvillat@ipn.mx

**Keywords:** enzyme inhibition, *Zea mays*, corn head smut, phytopathogenic fungus, starch

## Abstract

Corn head smut is a disease caused by the fungus *Sporisorium reilianum*. Chemical treatments and tolerant hybrids are available for control of this disease; however, these can lead to the development of resistant strains, complicating its management. This microorganism produces two extracellular enzymes—aspartyl protease EAP1 and xylanase SRXL1—which may be involved in the host penetration and colonization processes. Plants produce peptides that inhibit enzymes involved in phytopathogenesis, which could serve as tools to control plant pathogens. In this study, enzyme inhibitors were extracted from corn seed flours derived from two hybrids—a white variety (DK-2061) and a purple variety (BOGUI)—with the objective of evaluating their inhibitory effects on the enzymes EAP1 and SRXL1. Interestingly, the identified inhibitors were starches that showed 100% enzymatic inhibition. These compounds were characterized through Fourier transform infrared spectroscopy (FTIR), scanning electron microscopy (SEM), and energy-dispersive X-ray spectroscopy (EDS) analysis. The purified starches exhibited acetylation (1.52 ± 0.07% for DK-2061 and 1.16 ± 0.04% for BOGUI) as a result of the purification process, due to the use of an acetate regulator; however, they maintained their complete inhibitory activity against the studied enzymes. In contrast, the activity of the purified inhibitors was lost after incubation with α-amylase. Each isolated compound showed uncompetitive inhibition on both enzymatic activities, indicated by a decrease in *Km* and *V_max_* values, as determined using the Lineweaver-Burk equation. This represents the first report of the inhibitory effects of corn starches on aspartyl protease and xylanase extracted from *S. reilianum*. Therefore, these compounds could serve as valuable elements in strategies to manage head smut, potentially reducing the reliance on chemical fungicides.

## 1. Introduction

Corn (*Zea mays*) is one of the most widely produced cereals globally. While it is used primarily for human and animal consumption. It also serves as a raw material in various industries to create numerous products such as biofuels, sweeteners, oils, and beverages [1,2]. However, corn cultivation faces several phytosanitary challenges, such as the disease called corn head smut, which occurs worldwide. Corn head smut is characterized by the formation of dark carbonaceous masses on the head and cob [3] and is caused by the fungus *Sporisorium reilianum*. This basidiomycete has two phases: a haploid saprophytic phase and a dikaryotic parasitic phase [4,5]. As a natural inhabitant of the soil, it can survive for five to ten years in the form of teliospores that can be dispersed by wind, rain, and agricultural practices [5]. Under optimal temperature and humidity conditions, these spores can germinate in the young tissues of corn seedlings and form a basidium that produces the four basidiospores characterizing the haploid saprophytic phase [6]. These cells can fuse thanks to their sexual compatibility, and this action leads to the formation of the infective mycelial phase, which proliferates around the roots and subsequently penetrates and grows inside plant tissues; as such, the infection is systemic [6,7]. The initial symptoms of the disease typically appear after flowering, thus impacting grain production [5,7].

Methods for the control of this disease include cultural practices such as crop rotation, washing and disinfecting agricultural tools and machinery, eliminating diseased plants, and maintaining acceptable humidity levels throughout the early stages of crop growth [8]. These measures, however, require additional effort from producers. Another approach involves preventing contact between pathogens and hosts during the early stages of seedling development by means of chemical control, typically through applying systemic fungicides to the seeds [9,10]. Unfortunately, the chemicals used can generate pathogen resistance and can contaminate or damage agricultural soils [11]. Genetic improvement is another strategy employed based on the selection and evaluation of disease-tolerant hybrids [12,13]. While proven to be effective, this method has an important disadvantage because tolerance can switch to sensitivity from one year to the next [14]. Finally, the use of *Bacillus velezensis* as a biological control agent has shown promising results, with reports indicating that applying it in the field reduces the incidence of the disease and increases crop productivity [15,16].

Another way to manage phytopathogens is through enzyme inhibitors. Phytopathogens secrete a variety of enzymes that degrade plant cell walls, providing nutrients and facilitating the penetration and colonization of host tissues [17]. In response, plants produce enzyme inhibitors as part of their defense mechanisms [18,19,20]. Specific examples of these inhibitors include TAXI-I and TAXI-III (*Triticum aestivum* Xylanase Inhibitor), two xylanase inhibitors derived from wheat grains that effectively inhibit the activities of endoxylanases produced by pathogens such as *Fusarium graminearum*, *Botrytis cinerea*, and *Bipolaris sorokiniana* [21,22,23,24,25]. In addition, the polygalacturonase inhibitor protein (PGIP) obtained from beans has been shown to inhibit the polygalacturonase produced by *Fusarium moniliforme* FC-10 [26], while PGIP1 and PGIP2, also derived from beans, act against the polygalacturonase activity of *B. cinerea* [27,28]. The xylanase inhibitor-like protein (XILP), isolated from sorghum seeds, has also demonstrated antifungal effects against various pathogens, including *Mycosphaerella arachidicola*, *Fusarium oxysporum*, *Alternaria solani*, *Setosphaeria turcica*, *Pythium aphanidermatum*, *Verticillium dahliae*, and *Helminthosporium maydis* [29]. An XAIP-II inhibitor (Xylanase and α-amylase Inhibitor Protein), extracted from the bulbs of *Scadoxus multiflorus*, has in turn been identified to inhibit a GH11 xylanase produced by *Penicillium funiculosum* [30]. Finally, the rice xylanase inhibiting (RIXI) extracted from rice plants inhibits the GH11 xylanase produced by *Aspergillus niger* [31].

In a similar line, some protease inhibitors found in seeds, legumes, and plants play a significant role in controlling and inhibiting the proteolytic activities of phytopathogenic fungi and other organisms of medical interest [32,33,34]. Some of the earliest studies on these inhibitors focused on those isolated from soybeans, which were found to have the ability to inhibit trypsin [35,36]. Research has also identified protease inhibitors in pumpkins that are effective against pepsin and proteases secreted by the fungus *Glomerella cingulata* [37]. A Kunitz-type trypsin inhibitor called CaTI isolated from chickpea seeds has demonstrated 99% inhibitory activity against trypsin and the capacity to inhibit the growth of *Fusarium oxysporum* f.sp. *ciceris* [38]. Additionally, the inhibitor Merrtine, extracted from black soybean seeds, not only affects trypsin and chymotrypsin but also exhibits antifungal activities against *Alternaria alternata*, *F. oxysporum*, *Pythium aphanidermatum*, *Physalospora piricola*, *B. cinerea*, and *Fusarium solani*, among others [39] (Wang et al., 2014). Furthermore, a peptide found in potato has been shown to inhibit the proteolytic activities of trypsin, chymotrypsin, and papain while also suppressing the growth of the fungi *Candida albicans* and *Rhizoctonia solani* [40].

*S. reilianum* produces two extracellular enzymes: an aspartyl protease called EAP1 and a xylanase called SRXL1 [41,42]. These enzymes—similarly to those produced by other phytopathogenic fungi—may play significant roles in disease pathogenesis by facilitating degradation of the structural components of the plant cell wall [4].

This study involved isolating and characterizing enzyme inhibitors in aqueous extracts derived from seeds of two corn hybrids: a white variety (DK-2061) and a purple variety (BOGUI). The objective was to evaluate their effects on the enzymatic activities of aspartyl protease EAP1 and xylanase SRXL1 from *S. reilianum* in an effort to establish a foundation for developing alternative control methods for this devastating fungal pathogen.

## 2. Results

### 2.1. Enzyme Inhibitory Activity of the Aqueous Extracts Derived from Corn Seeds

The aqueous extracts from the seeds of both hybrid varieties of maize inhibited the enzymatic activity of both enzymes studied (Figure 1). In the case of the protease EAP1, the aqueous extract from the BOGUI hybrid displayed the highest inhibitory effect, with a value of 77.1 ± 1.6%, while DK-2061 showed an inhibition percentage of 70.6 ± 3.1% (Figure 1A). The xylanase SRXL1 was completely inhibited by both aqueous extracts tested (Figure 1B).

### 2.2. Purification of the Enzyme Inhibitors

During the purification process of the enzyme inhibitors by means of ion exchange chromatography, we found that the fractions showing 100% inhibition of the enzymes under study were from 17 to 23 for the DK-2061 hybrid; meanwhile, for BOGUI, they were from 24 to 30 (Appendix A).

### 2.3. Fourier Transform Infrared Analysis of the Enzyme Inhibitors

The inhibitors obtained were analyzed via Fourier-transform infrared spectroscopy (FT-IR) and compared to cornstarch and the flours and aqueous extracts obtained from the kernels (Figure 2). Analyses of the starch, flours, and aqueous extracts revealed specific spectral bands. The bands observed at 990–994 cm^−1^ and 1411–1417 cm^−1^ correspond to the C-OH and C-O-C bonds associated with the tensile vibrational mode and the bending vibrational mode of the anhydroglucose ring, respectively. A bending vibration associated with the glycosidic bond was observed in the region between 1075 and 1077 cm^−1^. The asymmetric stretching vibrational mode, observed between 1147 and 1149 cm^−1^, corresponds to the C-O-C bond of glucose, while the region between 1638 and 1644 cm^−1^ corresponds to a bending vibration attributed to H-O-H interactions, possibly due to adsorbed water.

A band identified in the range of 2923–2926 cm^−1^ corresponds to the asymmetric stretching of the C-H bond associated with glucose. Additionally, the presence of hydroxyl (OH) groups—characteristic of glucose—was observed in the 3277–3284 cm^−1^ region. These vibrations are typical of starch and were detected in the flours and aqueous extracts derived from both hybrids (Figure 2A,B). The purified inhibitors exhibited shifts in some of the characteristic bands of starch. The band around 990 cm^−1^ shifted to 1013 cm^−1^ and 1014 cm^−1^ for the DK-20261 and BOGUI hybrids, respectively, possibly due to structural changes resulting from the acetylation process. The band around 1070 cm^−1^ was observed only in the inhibitor derived from the DK-2061 hybrid (Figure 2A). Both inhibitors showed a band in the 2929–2932 cm^−1^ range, corresponding to the C-H bond of glucose. The asymmetric bands around 1400 cm^−1^ and 1560 cm^−1^ indicate the interaction of the inhibitor with the acetate regulator. The bands around 1739 cm^−1^ for DK-2061 and 1731 cm^−1^ for BOGUI correspond to the C=O group of the acetyl ester functional group in the inhibitors derived from the purification process.

### 2.4. Scanning Electron Microscopy of Enzyme Inhibitors

Scanning electron microscopy (SEM) was performed to characterize the purified enzyme inhibitors by comparing them to the flours and aqueous extracts obtained from each hybrid studied. The flours generally presented oval, spherical, and elongated structures (Figure 3A and Figure 4A)—a morphology that coincides with the typical shape of starch granules and, hence, supports the results obtained via FT-IR (Figure 2). Starch granules were observed to remain intact in the aqueous extracts (Figure 3B and Figure 4B). Upon analyzing the micrographs of the purified inhibitors, marked changes in the surface of the granules were identified. The inhibitor obtained from DK-2061 exhibited a porous appearance (Figure 3C,D). Although the inhibitor derived from the BOGUI hybrid also displayed some porosity, it was to a lesser extent (Figure 4C,D). These observations may be linked to the acetylation of starch, which aligns with the results obtained from the FT-IR analysis (Figure 2). In the case of the purified inhibitor from the BOGUI hybrid, some crystals of NaCl from the regulators used in the purification process were also observed (Figure 4C,D).

### 2.5. Energy Dispersive Spectroscopy Analysis of the Purified Inhibitors

Energy Dispersive Spectroscopy (EDS) analysis revealed that the elemental composition of the flours, aqueous extracts, and purified inhibitors from both hybrids consisted primarily of carbon and oxygen. In some cases, small amounts of phosphorus, nitrogen, and aluminum were detected. Sodium and chlorine were also present in the purified inhibitors (Appendix A).

### 2.6. Qualitative Analysis of the Starch

To corroborate the chemical nature of the flours, the aqueous extracts and purified inhibitors were analyzed qualitatively to determine whether they corresponded to starch. The tests were positive in all cases (Appendix A).

### 2.7. Determination of the Degree of Acetylation of the Purified Inhibitors

Based on the results of the FT-IR and SEM analyses, the degree of acetylation of the purified inhibitors was determined. The inhibitor from the white hybrid DK-2061 showed a value of 1.52 ± 0.07%, higher than the 1.16 ± 0.04% observed for the purple BOGUI hybrid. Regarding the degree of substitution, the values were 0.06 ± 0.003% for DK-2061 and 0.04 ± 0.002% for BOGUI. In both cases, the differences were statistically significant.

### 2.8. Activity of α-Amylase on Flours and Purified Inhibitors from Corn Hybrids DK-2061 and BOGUI

It was determined that the purified flours and inhibitors from the corn hybrids DK-2061 and BOGUI can act as substrates for the α-amylase enzyme (Table 1).

### 2.9. Effect of α-Amylase on the Stability of Inhibitors Obtained from Corn Hybrids DK-2061 and BOGUI

It was observed that the flours and purified inhibitors derived from the corn hybrids DK-2061 and BOGUI lost their ability to inhibit the EAP1 and SRXL1 enzymes after being incubated with α-amylase (Table 2), confirming that the substances responsible for inhibition are starches.

### 2.10. Effects of the Inhibitors on the Kinetic Parameters of EAP1 and SRXL1

To assess the effects of the inhibitors on the kinetic parameters (i.e., *Km* and *V_max_*) of the enzymes studied, experiments were conducted to determine the concentration of each purified inhibitor that was required to achieve 50% inhibition. The results showed that for the protease, 20 µg of the DK-2061 hybrid inhibitor resulted in 51.7 ± 1.8% inhibition, while 30 µg of the BOGUI corn inhibitor produced 52.0 ± 1.8% inhibition (Table 3). For xylanase, 5 µg of inhibitor caused 50.5 ± 0.9% and 54.4 ± 0.8% inhibition for DK-2061 and BOGUI, respectively (Table 3).

Based on these results, two concentrations of the purified inhibitors were selected to evaluate their effects on the enzymatic kinetics of the two activities under study. We found that the *Km* and *V_max_* values for both the aspartyl protease EAP1 and the xylanase SRXL1 decreased in the presence of the inhibitor, as did the catalytic efficiency (*V_max_*/*Km*). These findings demonstrate that the inhibition established is of the uncompetitive type (Table 4). Figure 5 and Figure 6 present graphs of the enzymatic kinetics determined using the Lineweaver–Burk method.

### 2.11. Effect of the Purified Enzyme Inhibitors on the Development of S. reilianum

The effects of distinct concentrations of the purified enzyme inhibitors on the development of *S. reilianum* were tested; however, the results showed that these compounds had no antifungal activity. In particular, yeast-like growth spread across the entire surface of the medium on the plate was observed.

## 3. Discussion

Plants have developed protection or resistance mechanisms against phytopathogenic attacks [43,44] through intracellular signals that induce the synthesis of defense compounds [45,46,47]. These include enzyme inhibitors, which block the activity of enzymes produced by pathogenic agents [46]. In this study, enzyme inhibitors were identified in aqueous extracts from the seeds of two corn hybrids—one white (DK-2061) and one purple (BOGUI)—which showed inhibitory effects on the aspartyl protease EAP1 and xylanase SRXL1 enzymes of *S. reilianum*. A bifunctional inhibitor produced by the bacterium *Bacillus* sp. Called ATBI (Alkalophilic Thermophilic *Bacillus* Inhibitor), it has been described as having inhibitory activity against xylanases and proteases produced by the fungi *Trichoderma reesei* and *Aspergillus oryzae*; however, unlike the inhibitor in this study, ATBI is a peptide that exhibits competitive inhibition effects on the aforementioned enzymes [48].

The reports available to date on enzyme inhibitors from plants that act on xylanases and proteases show that they are proteins [32,49,50]. In transgenic plants, the expression of these compounds confers resistance to fungal diseases [31,38,51,52,53]. Moreover, there have been reports that several of these substances, isolated from various seed types, not only inhibit enzymatic activity but also exert antifungal effects on numerous phytopathogens [29,39,54,55,56,57,58,59,60,61]. Two protein inhibitors that act on α-amylase and trypsin have been isolated from corn seeds [60,61], one of which was also shown to exhibit antifungal activity [60]. In this study, the inhibitor isolated from corn seeds is a starch that did not exhibit antifungal activity against *S. reilianum*. This polymer is one of the most abundant polysaccharides in nature, which can be found in the seeds of various cereals, fruits, and legumes. It is composed of linear glucose chains that form amylose with branches that form amylopectin [62]. To date, no enzyme inhibition activities have been attributed to this carbohydrate. In our work, it was found to present an uncompetitive mode of inhibition in which the inhibitor binds to the enzyme–substrate complex to distort the structure of the active site and cause decreases in the *Km* and *V_max_* values [63].

During the purification process, we observed that the starch underwent a structural change. This was confirmed via FTIR analysis, which revealed a characteristic band indicating the presence of ester groups that can be attributed to acetylation due to the use of an acetate regulator during purification. In this case, the hydroxyl groups are replaced by the carbonyl groups of the ester. These results are similar to those reported by Daza and Parra [64]. The modification of starch involves reactions that can occur with the three hydroxyl groups of glucose [65,66], which is reflected in the degree of substitution. In this study, these values for the purified starches were low; however, the appearance of the granules was porous, distinct from what was observed in the flours. A low degree of substitution—ranging from 0.01 to 0.2—suggests potential for use in various applications, including as an adhesive, thickener, stabilizer, and texturizer, as well as in film formation [67,68].

Regarding morphological characteristics, starch is organized into granules that range in diameter from 1 to 100 μm, usually with a spherical, oval, or elongated shape. The distribution can be either unimodal or bimodal and, depending on its associations, can be observed individually or in groups [62]. Flours from the DK-2061 and BOGUI hybrids generally exhibited these structures. As mentioned above, the purified starches showed a certain degree of porosity. Some studies have reported that an increase in this characteristic can favor an increase in the surface area [68].

Peptide inhibitors can inhibit the development of plant pathogens [32,49,50]. This study revealed that the purified inhibitors had no antifungal effect on *S. reilianum*. Although starch alone does not possess antimicrobial activity, other agents can be added to this polymer to confer this property [69].

This study is the first report that starch extracted from the maize seeds of two hybrids—one white (DK-2061) and one purple (BOGUI)—inhibits the enzymes aspartyl protease EAP1 and xylanase SRXL1 from *S. reilianum*. Although these findings were obtained in vitro, they may lay the groundwork for the development of strategies to control corn head smut by blocking the extracellular activities involved in plant cell wall degradation. Future research could focus on testing starches from the corn hybrids evaluated in this study to control corn head smut in plants, as well as exploring potential synergistic effects when combining these inhibitors with other disease control methods.

## 4. Materials and Methods

### 4.1. Microorganisms and Conservation

In this study, a haploid strain of *S. reilianum* was isolated from teliospores collected from diseased corn plants in a state of physiological maturity, located in the Mezquital Valley, Hidalgo, Mexico. This strain was generously provided by Dr. Santos Gerardo Leyva Mir of the Autonomous University of Chapingo, Mexico. To ensure its preservation, the strain was cultured in inclined tubes containing YEPD medium (1% yeast extract, 2% peptone, 2% glucose, and 2% agar), supplemented with 1 µg/mL of streptomycin. The tubes were incubated at 28 °C for 72 h, then sterile mineral oil was added to cover the microbial growth. The prepared tubes were stored at room temperature [15].

### 4.2. Obtaining Aqueous Extracts from Corn Seeds

Two hybrid varieties of corn seeds were utilized, one of which is known to be tolerant to corn smut (hybrid DK-2061, DeKalb^®^, Bayer, IL, USA), and the other from a purple variety (BOGUI, Biosuva, Mixquiahuala, Hidalgo, Mexico). The seeds were pretreated with a 10% sodium hypochlorite solution for 15 min. The kernels were then washed four times in sterile distilled water to eliminate the chemical treatment that covered them and left to dry for 96 h in a dryer at room temperature and 19% humidity. At that point, they were ground in a mill to obtain a fine flour, which was sifted through a #10 sieve with 2 mm holes. Distilled water was added at a ratio of 1:4, and the solution was filtered to eliminate the largest particles. The aqueous extract was stored at −4 °C.

### 4.3. Obtaining the Crude Enzyme Extracts with Aspartyl Protease EAP1 and Xylanase SRXL1 Activity

The methodologies described by Mandujano-González et al. [41] and Álvarez-Cervantes et al. [42] were followed to obtain the crude enzymatic extracts with EAP1 and SRLX1 activity. The composition of the production media for protease included 1.7 g/L yeast nitrogen base (without amino acids or ammonium sulfate), 20 g/L glucose, and 5 g/L (NH_4_)_2_SO_4_. For xylanase production, the media contained 0.6 g/L KH_2_PO_4_, 0.5 g/L MgSO_4_·7H_2_O, 0.4 g/L K_2_HPO_4_, 0.05 g/L FeSO_4_·7H_2_O, 0.05 g/L MnSO_4_·H_2_O, 0.001 g/L ZnSO_4_·7H_2_O, and 0.5 g/L birch xylan. Two pre-inocula were prepared—namely, at 24 h for EAP1 and 48 h for SRXL1—for which the biomass of the complete development of *S. reilianum* was taken from a plate with YEPD and inoculated into 250 mL flasks with 50 mL of each of the aforementioned media and incubated at 28 °C with agitation at 150 rpm. The necessary amounts of pre-inocula were taken to adjust 50 mL of each medium in 250 mL flasks to an absorbance of 0.2 at 600 nm. Incubation was carried out under the aforementioned conditions for 96 h. The biomass was then separated via centrifugation at 8000 rpm. The supernatant constituted the crude enzyme extract (CEE), which was collected and stored in 10 mL aliquots at −4 °C for up to eight days.

### 4.4. Determination of the Activities of the Aspartyl Protease EAP1 and the Xylanase SRXL1

To assess the enzymatic activity of the EAP1 protease, 400 μL of a substrate solution (0.5% bovine serum albumin in 0.05 M of acetate buffer at pH 3) was mixed with 100 μL of the CEE. This mixture was then incubated for 120 min at 37 °C. The reaction was terminated by adding 500 μL of 10% trichloroacetic acid. Afterward, the mixture was centrifuged at 13,000 rpm for 5 min. An enzymatic reaction control was included, which was stopped at time “zero”, together with a negative control in which the CEE was replaced with distilled water. The supernatant was then collected to quantify the peptides released using the modified Lowry method. For this quantification, 200 μL of the supernatant were mixed with 200 μL of reagent A prepared using 500 μL of 1% CuSO_4_, 500 μL of 2.7% KNaC_4_H_4_O_6_, 5 mL of 24% Na_2_CO_3_, and 6 mL of 1 M NaOH. This solution was incubated at room temperature for 10 min. Next, 1 mL of reagent B (Folin–Ciocalteu, diluted in water at a ratio of 1:20) was added, and the mixture was incubated under darkness at 37 °C for 30 min. Absorbance was measured at 660 nm using a tyrosine standard curve as a reference. One unit of enzyme activity was defined as the amount of enzyme required to release 1 μmol of tyrosine per minute under these specific assay conditions.

The activity of xylanase SRXL1 was determined by measuring the release of reducing sugars via the Miller method [70]. The enzymatic reaction consisted of 450 μL of 0.3% birch xylan in a 0.1 M acetate buffer at pH 5.3 as the substrate and 50 μL of the CEE. The mixture was incubated at 50 °C for 5 min. Then, 1 mL of DNS reagent was added, which contained the following components: NaOH (10 g/L), 3,5-dinitrosalicylic acid (10 g/L), KNaC_4_H_4_O_6_ (2 g/L), Na_2_SO_3_ (0.5 g/L), and phenol (2 g/L). An enzymatic reaction control was included, where the DNS was added at time “zero”, together with a negative control in which the CEE was replaced with distilled water. The reaction was stopped by placing the tubes in a boiling water bath for 5 min, after which they were cooled in an ice bath. Absorbance was measured at 576 nm. The amount of reducing sugars released was calculated using a standard curve generated from a xylose solution. One unit of enzyme activity was defined as the amount of enzyme required to release 1 μmol of xylose per minute under these specific assay conditions.

### 4.5. Enzyme Inhibition Assays

Inhibitory activity was determined by mixing 50 μL of the aqueous extracts obtained from the seeds with 100 μL and 50 μL of the CEEs derived from EAP1 and SRXL1, respectively. The mixture was incubated for 5 min at 37 °C, following which the enzymatic determinations were conducted as described previously. Distilled water was used as a control instead of the aqueous extract. All experiments were conducted in triplicate. The results are expressed as percentages of inhibition.

### 4.6. Purification of the Enzyme Inhibitor

In this step, 50 g of seeds from each hybrid (i.e., DK-2061 and BOGUI) were washed, as described previously, dried for 96 h, and ground into a flour using a mill. The methodology described by Fierens et al. [23] was followed for the purification process in which 40 mL of a 0.1% L-ascorbic acid solution was added to 10 g of each flour. The mixture was left to stand at 4 °C for 18 h, then centrifuged at 15,000 rpm for 30 min at 4 °C. The precipitate was discarded, and the supernatant was treated with an amount of CaCl_2_ required to achieve a final concentration of 2 g/L. The pH was adjusted to 8.5 with 2 M of NaOH. The mixture was left at 4 °C for another 18 h before centrifuging again under the same conditions.

At that point, the supernatant was adjusted to pH 4.5 by adding 2 M HCl. It was then evaluated for its inhibitory activity against the enzymes under study. The extract obtained was filtered through a 0.22 μm pore nylon membrane, and the resulting filtrate was passed through a 5 mL Q Sepharose HP XK50 anion exchange column (Cytiva, Marlborough, MA, USA) coupled to an FPLC system (ӒKTA pure™ brand Fast Protein Liquid Chromatography, Cytiva, Marlborough, MA, USA). The column was equilibrated with 25 mM of acetate buffer, pH 4.5. The fractions were eluted using a gradient of 0 to 1 M of NaCl in 25 mM of acetate buffer, pH 4.5, with a flow rate of 1 mL/min and assayed for inhibitory activity against the EAP1 and SRXL1 enzymes.

All fractions that exhibited enzyme inhibitory activity were collected in vials and lyophilized using the BIOBASE BK-FD10P^®^ system (BIOBASE Group, Jinan, China), maintaining a constant pressure of 5 kPa and a temperature of −50 °C for 96 h. Upon completing the lyophilization process, the vials were sealed and stored at room temperature.

### 4.7. Purification of the Aspartyl Protease EAP1 and Xylanase SRXL1 Enzymes

Purification of enzymes was performed according to the methods established by Mandujano-González et al. [41] for aspartyl protease EAP1 and by Álvarez-Cervantes et al. [42] for xylanase SRXL1.

### 4.8. Fourier-Transform Infrared Spectroscopy (FT-IR) Analysis

The purified inhibitors, grain flour, and lyophilized aqueous extracts from each hybrid were analyzed via Fourier-transform infrared spectroscopy using the AGILENT CARY 630^®^ FT-IR system (Agilent Technologies, Santa Clara, CA, USA). Spectra were recorded over a range of 4000 to 400 cm^−1^, with an average of 100 scans conducted at a resolution of 8 cm^−1^. Comparisons were made with respect to commercial corn starch (Meyer 5430, Sigma-Aldrich, St. Louis, MO, USA).

### 4.9. Scanning Electron Microscopy (SEM) and Energy-Dispersive X-Ray Spectroscopy (EDS) Analyses

The appearance and changes generated during the purification of the inhibitor were observed under a QUANTA FEG 250^®^ scanning electron microscope (SEM) (Thermo Fisher Scientific, Waltham, MA, USA) at 5 kV and a pressure of 6.04 × 10^−1^. This examination included the untreated flours and the lyophilized aqueous extract from each hybrid. The samples were fixed to a graphite conductive tape mounted on sample holders to facilitate visualization of the surfaces of the materials. Scanning electron micrographs were recorded at magnifications of 1000×, 2500×, and 5000×.

In addition, a semiquantitative EDS analysis was performed using the BRUKER XFlash 6160^®^ detector (Bruker Corporation, Billerica, MA, USA) coupled to the SEM, also at 5 kV and a pressure of 6.04 × 10^−1^.

### 4.10. Qualitative Determination of Starch

For the qualitative determination of starch, 10 mg of the flour samples, aqueous extracts, and lyophilized purified inhibitors were mixed with 0.9 mL of distilled water and 100 µL of an I_2_/KI solution (20 mg of iodine and 200 mg of potassium iodide in 10 mL) in order to achieve a final concentration in the reaction mixture of 0.2 mg of I_2_ and 2 mg of KI. A positive test result was indicated by the development of a blue color [71].

### 4.11. Determination of the Degrees of Acetylation and Substitution of the Purified Inhibitors

The methodology described by Zhang et al. [72] was followed, making some modifications, to determine the degrees of acetylation and substitution of the purified inhibitors. In this case, 25 mg of inhibitor was dissolved in 0.625 mL of distilled water, and the resulting solution was placed in a 25 mL Erlenmeyer flask, to which five drops of phenolphthalein indicator were added. After that, the necessary amount of 0.1 M NaOH was added until the solution turned pink—i.e., the phenolphthalein endpoint—followed by stirring for 5 min. Subsequently, 0.125 mL of 0.5 M NaOH was added, and the solution was stirred again for 30 min, and a titration was performed with 0.5 M HCl. The volume of the acid consumed was recorded as V_1_. The same procedure was performed with the corn flour from which the inhibitor was purified as a control. In this case, the volume of HCl consumed in the titration corresponded to the V_0_ value. The following formula was used to calculate the degree of acetylation:$$A\;=\;\frac{\left(V_{1}\;-\;V_{0}\right)\;\times \;C\;\times \;0.043\;\times \;100}{m}$$
where A represents the degree of acetylation as a percentage (%); V_0_ is the volume of HCl used in the titration of the control (in mL); V_1_ is the volume of HCl used in the titration of the sample (in mL); C is the concentration of HCl; m is the mass of the purified inhibitor (in mg); and 0.043 corresponds to the milliequivalent of the acetyl group [73].

The degree of substitution (DS) in the acetylated starch represents the average number of hydroxyl groups replaced by acetyl groups in the glucose anhydride unit, calculated using the following formula:$$DS\;=\;\frac{162A}{\left(4300\;-\;42A\right)}$$
where A represents the degree of acetylation expressed as a percentage (%). The number 162 refers to the molecular weight of the anhydroglucose unit, while 42 corresponds to the molecular weight of acetate, minus one. The quantity of 4300 was achieved by multiplying 100 g of starch by the molecular weight of the acetyl group [72].

### 4.12. Effects of α-Amylase on Flours and Purified Inhibitors from Corn Hybrids DK-2061 and BOGUI

The activity of α-amylase was measured according to the release of reducing sugars using the Miller method [70]. In this assay, flours and purified inhibitors served as substrates. Potato starch was also included as a control.

The enzymatic reactions consisted of 75 mg of the flours and 30 mg of the purified inhibitors in 250 µL of 20 mM sodium phosphate buffer and 6.7 mM NaCl (pH 6.9). Next, 5 µL of the *B. amyloliquefaciens* α-amylase enzyme (Sigma-Aldrich, A7595, Sigma-Aldrich, St. Louis, MO, USA) was added to this mixture and incubated for 10 min at 50 °C, following which 1 mL of DNS reagent was added. An enzymatic reaction control was included, where the DNS was added at time “zero”, together with a negative control in which the CEE was replaced with distilled water. The reaction was stopped by placing the tubes in a boiling water bath for 10 min. Absorbance was measured at 576 nm. The amount of reducing sugars released was calculated using a standard curve generated from a glucose solution. One unit of enzyme activity was defined as the amount of enzyme required to release 1 μmol of glucose per minute under these specific assay conditions. This experiment was also conducted in triplicate.

### 4.13. Effects of α-Amylase on the Stability of Inhibitors Obtained from Corn Hybrids DK-2061 and BOGUI

To evaluate the effects of α-amylase on the purified inhibitors, the following procedure was used: 75 mg of the flours and 30 mg of the purified inhibitors were mixed with 250 μL of 20 mM sodium phosphate buffer (pH 6.9) and 5 μL of the enzyme, then incubated for 10 min at 50 °C. The resulting solution was used to measure its inhibitory effect on EAP1 and SRXL1, as described in Section 4.5. The following controls were used: Control 1, where the enzyme was denatured in a boiling water bath for 10 min, and Control 2, where distilled water was used instead of the enzyme. This experiment was also conducted in triplicate.

### 4.14. Effects of the Inhibitors on the Kinetic Parameters of Aspartyl Proteases EAP1 and Xylanase SRXL1

Different concentrations of inhibitors obtained from each hybrid were chosen to assess their effects on the kinetic parameters *Km* and *V_max_* of the enzymes under study, based on the observed IC50 values. In this case, 0, 18, and 23 µg from the DK2061 hybrid and 0, 24, and 30 µg from the BOGUI hybrid were used for EAP1. For SRXL1, 0, 2.5, and 7.5 µg from DK2061 and 0, 2.5, and 4.0 µg from BOGUI were employed. Enzymatic reactions were conducted in the presence and absence of the purified inhibitor, as described previously. The substrate concentrations were as follows: for the protease, albumin was used at concentrations ranging from 0.3 to 6 mg/mL in a 0.05 M citrate buffer at pH 3.2. For xylanolytic activity, birch xylan was utilized in a 0.1 M acetate buffer at pH 5.3, with concentrations of 0.5 to 3 µg/mL.

To determine the type of inhibition established between the purified inhibitors and the enzymes of interest, double reciprocal Lineweaver-Burk plots (1/V versus 1/S) were constructed for each inhibitor concentration. The “y” intercept corresponds to the inverse of *V_max_*, while the “x” intercept represents −1/*Km*. Both *Km* and *V_max_* were determined experimentally using the following equation:$$\frac{1}{V}\;=\;\frac{Km}{V_{max}}\;\times \;\frac{1}{S}\;+\;\frac{1}{V_{max}}$$
where *Km* represents the Michaelis–Menten affinity constant; V indicates the reaction rate, *V_max_* denotes the maximum reaction rate; and S refers to the substrate concentration [74]. The type of inhibition was determined by comparing *Km*, *V_max_*, and the *Km*/*V_max_* ratio in the presence and absence of the inhibitor. All assays were performed in triplicate.

### 4.15. Effects of the Purified Enzyme Inhibitors on the Development of S. reilianum

A pre-inoculum of *S. reilianum* was prepared by harvesting the completely grown fungus from a plate containing YEPD medium and then inoculating it in a 250 mL flask with 50 mL of YEPD broth. The flask was incubated at 28 °C with agitation at 150 rpm for 24 h. The necessary inoculum was taken from this culture to adjust 50 mL of the same medium to an absorbance of 0.2 at 600 nm, after which it was incubated under the same conditions. From the resulting culture, 200 μL was transferred to test tubes containing 5 mL of a semi-solid medium made of molten YEPD (1% yeast extract, 2% peptone, 2% glucose, and 0.5% agar). Distinct concentrations of the purified inhibitor (25, 50, 100, 200, 300, 400, and 600 μg) in acetate buffer (pH 4.5, 25 mM), sterilized by filtration, were then added to the test tubes. The mixture was shaken and spread on the surface of 60 mm diameter Petri dishes containing YEPD agar, which were then incubated at 28 °C for 96 h. For the control group, sterile acetate buffer was added instead of the inhibitor. At the end of the incubation period, the microbial growth distributed over the dishes was examined visually. This experiment was also conducted in triplicate.

### 4.16. Statistical Analyses

Statistical analyses were conducted using a one-way ANOVA with multivariate comparisons via a Tukey–Kramer test at a 95% confidence level (*p* ≤ 0.05). The data were tested for normal distribution and homoscedasticity via the Shapiro–Wilk or Kolmogorov–Smirnov and Levene’s tests, respectively. The NCSS 12 data analysis software was utilized.

## Figures and Tables

**Figure 1 ijms-26-09974-f001:**
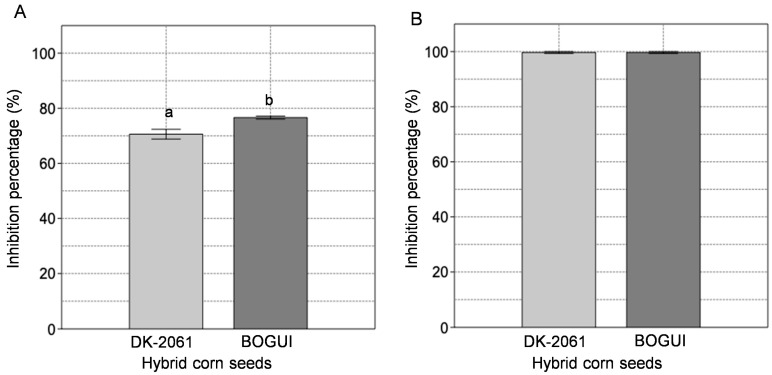
Percentage inhibition of the activity of the aspartyl protease EAP1 (**A**) and xylanase SRXL1 (**B**) enzymes of *S. reilianum* by aqueous extracts obtained from hybrid corn seeds (white hybrid DK-2061, purple hybrid BOGUI). Letters specify the results of the statistical analyses. Different letters indicate a statistically significant difference with a *p*-value < 0.05. The results of the statistical analysis are available in Appendix A. The determinations were performed in triplicate. The error bars show the standard deviation.

**Figure 2 ijms-26-09974-f002:**
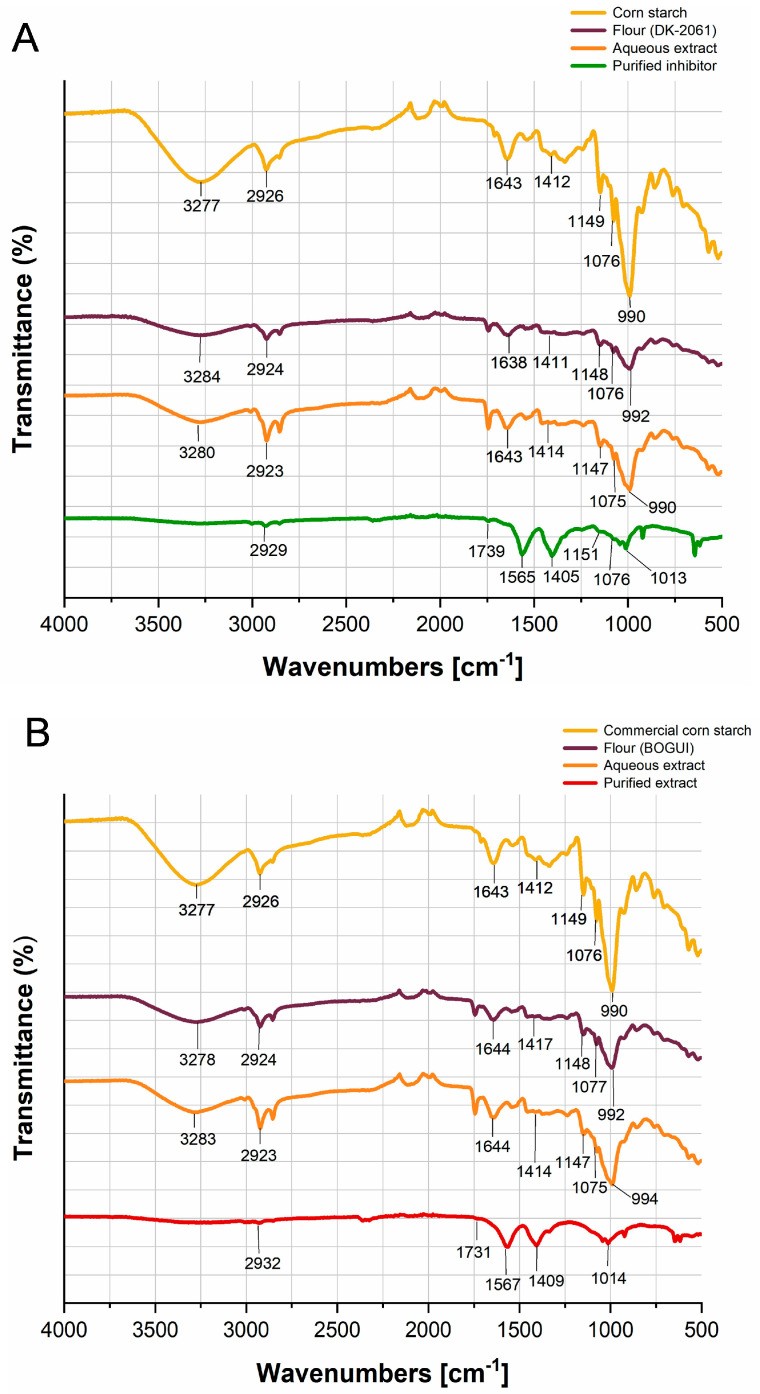
FT-IR spectra of the purified inhibitors with activity against the aspartyl protease EAP1 and xylanase SRXL1 enzymes of *S. reilianum*, and of the flours and aqueous extracts obtained from the two corn hybrids. The spectra of the starches from these grains are also shown. (**A**) White DK-2061 hybrid; (**B**) purple BOGUI hybrid. Spectra were recorded over a range of 4000 to 400 cm^−1^, with an average of 100 scans and conducted at a resolution of 8 cm^−1^.

**Figure 3 ijms-26-09974-f003:**
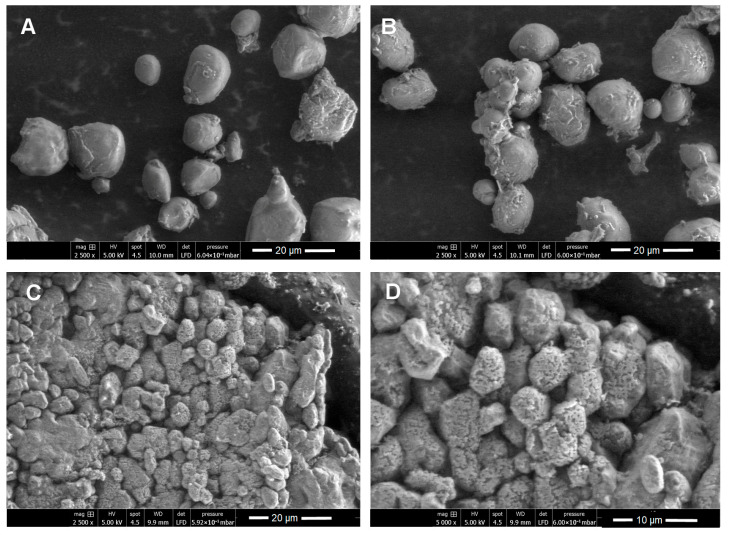
SEM images of flour (**A**), the aqueous extract (**B**), and the purified inhibitor (**C**,**D**) with activity against the aspartyl protease EAP1 and xylanase SRXL1 enzymes of *S. reilianum*, obtained from maize seeds of the white DK-2061 hybrid. Observations in (**A**–**C**) were made at 2000×, and that in (**D**) at 5000×.

**Figure 4 ijms-26-09974-f004:**
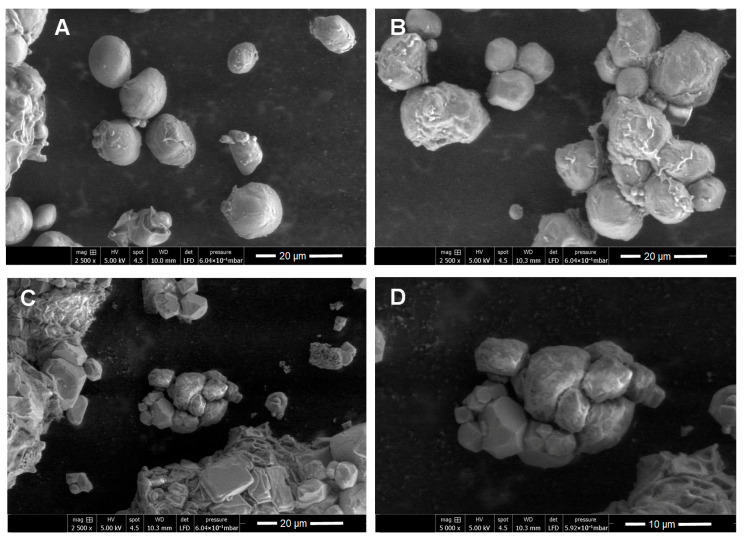
SEM images of flour (**A**), the aqueous extract (**B**), and the purified inhibitor (**C**,**D**) with activity against the aspartyl protease EAP1 and xylanase SRXL1 enzymes of *S. reilianum*, obtained from maize seeds of the purple BOGUI hybrid. Observations in (**A**–**C**) were made at 2000×, and that in (**D**) at 5000×.

**Figure 5 ijms-26-09974-f005:**
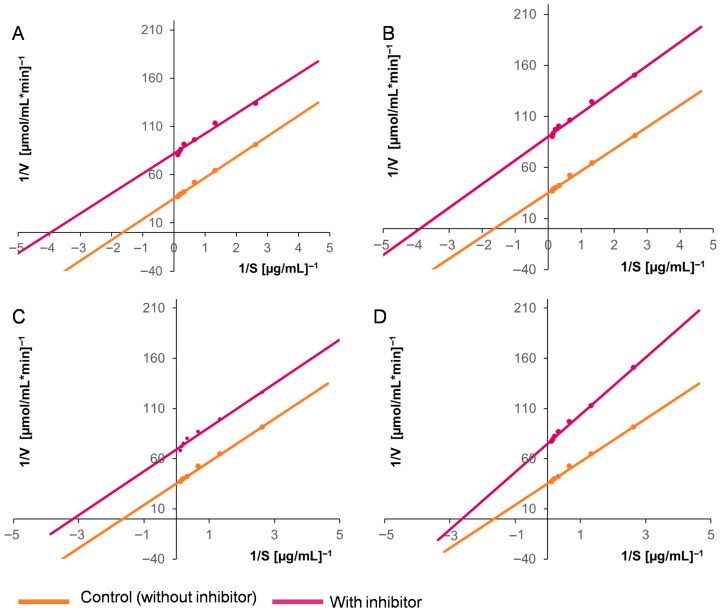
Lineweaver–Burk plots showing the effects of the purified inhibitors from maize seeds on the activity of *S. reilianum* aspartyl protease EAP1: (**A**) 18 µg and (**B**) 23 µg of the purified inhibitor from the white DK-2061 hybrid: (**C**) 24 µg and (**D**) 30 µg of the purified inhibitor from the purple BOGUI hybrid. The determinations were performed in triplicate.

**Figure 6 ijms-26-09974-f006:**
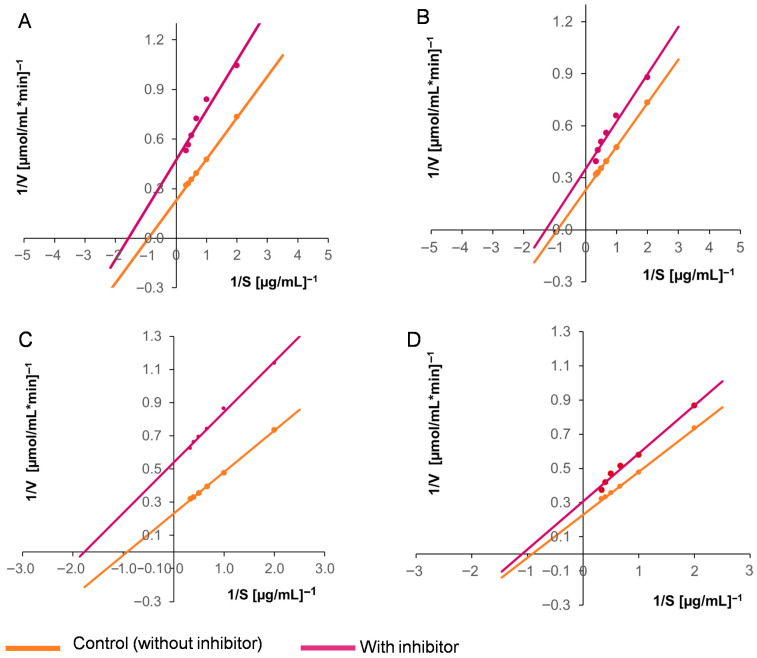
Lineweaver–Burk plots showing the effects of the purified inhibitors from maize seeds on the activity of *S. reilianum* xylanase SRXL1: (**A**) 2.5 µg and (**B**) 7.5 µg of the purified inhibitor from the white DK-2061 hybrid: (**C**) 2.5 µg and (**D**) 4.0 µg of the purified inhibitor from the purple BOGUI hybrid. The determinations were performed in triplicate.

**Table 1 ijms-26-09974-t001:** Activity of *α*-amylase from on flours and purified inhibitors derived from the corn hybrids DK-2061 and BOGUI.

Substrate	α-Amylase Activity (U/mL)
Potato starch	47.4 ± 1.2 ^A^
Flour from hybrid DK-2061	31.8 ± 0.3 ^D^
Flour from hybrid BOGUI	39.5 ± 0.6 ^B^
Purified inhibitor from DK-2061	22.7 ± 0.2 ^E^
Purified inhibitor from BOGUI	32.1 ± 4.3 ^C^

The determinations were performed in triplicate. Capital letters show the results of statistical analyses, with identical letters between treatments indicating no statistically significant differences (*p* ≤ 0.05). The results of the statistical analysis are available in Appendix A.

**Table 2 ijms-26-09974-t002:** Effect of α-amylase on the stability of inhibitors obtained from corn hybrids DK-2061 and BOGUI.

Substrate	% Inhibition					
	Treatment		Control 1		Control 2	
	EAP1	SRXLI	EAP1	SRXLI	EAP1	SRXLI
Flour from DK-2061	0 ± 0	0 ± 0	100 ± 0	100 ± 0	100 ± 0	100 ± 0
Flour from BOGUI	0 ± 0	0 ± 0	100 ± 0	100 ± 0	100 ± 0	100 ± 0
Purified inhibitor from DK-2061	0 ± 0	0 ± 0	100 ± 0	100 ± 0	100 ± 0	100 ± 0
Purified inhibitor from BOGUI	0 ± 0	0 ± 0	100 ± 0	100 ± 0	100 ± 0	100 ± 0

The determinations were performed in triplicate.

**Table 3 ijms-26-09974-t003:** Determination of IC50 values for inhibitors obtained from seeds of the corn hybrids DK-2061 and BOGUI on the aspartyl protease EAP1 and the xylanase SRXL1 of *S. reilianum*.

Aspartyl Protease EAP1				Xylanase SRXL1			
DK-2061		BOGUI		DK-2061		BOGUI	
µg	% Inhibition	µg	% Inhibition	µg	% Inhibition	µg	% Inhibition
0.0	0.0 ± 0.0 ^A^	0.0	0.0 ± 0.0 ^A^	0.0	0.0 ± 0.0 ^A^	0.0	0.0 ± 0.0 ^A^
5.0	11.9 ± 0.6 ^B^	10.0	19.9 ± 0.8 ^B^	2.5	46.4 ± 0.9 ^B^	2.5	42.4 ± 1.0 ^B^
10.0	14.5 ± 1.5 ^B^	15.0	26.1 ± 0.6 ^C^	5.0	50.5 ± 0.9 ^C^	5.0	54.4 ± 0.8 ^C^
15.0	17.1 ± 1.7 ^B^	20.0	29.2 ± 0.5 ^D^	10.0	54.8 ± 0.6 ^D^	7.5	64.3 ± 0.7 ^D^
20.0	51.7 ± 0.2 ^C^	25.0	34.5 ± 1.1 ^E^	15.0	57.5 ± 0.6 ^E^	10.0	96.7 ± 0.1 ^E^
25.0	92.6 ± 0.2 ^D^	30.0	52.0 ± 1.8 ^F^	20.0	65.6 ± 0.4 ^F^	-	-
-	-	35.0	92.1 ± 0.2 ^G^	25.0	96.6 ± 0.1 ^G^	-	-

Capital letters show the results of statistical analyses, with identical letters between treatments indicating no statistically significant differences (*p* ≤ 0.05). The results of the statistical analysis are available in Appendix A. The determinations were performed in triplicate.

**Table 4 ijms-26-09974-t004:** Effect of enzyme inhibitors obtained from corn seeds of the DK-2061 and BOGUI hybrids on the kinetic parameters of the aspartyl protease EAP1 and of the xylanase SRXL1 of *S. relianum*.

Inhibitor	Inhibitor Concentration (µg/mL)	*V_max_*	*Km*	*V_max_/Km*	R^2^ Value	Type of Inhibition
Aspartyl protease EAP1						
DK-2061	No inhibitor	0.028 ± 0.001 ^A^	0.61 ± 0.028 ^A^	0.050	0.996	Uncompetitive
	15	0.012 ± 0.001 ^B^	0.25 ± 0.013 ^B^	0.048	0.979	
	20	0.011 ± 0.001 ^C^	0.26 ± 0.020 ^B^	0.044	0.989	
BOGUI	No inhibitor	0.028 ± 0.001 ^A^	0.61 ± 0.028 ^A^	0.050	0.996	Uncompetitive
	25	0.014 ± 0.001 ^B^	0.32 ± 0.021 ^B^	0.046	0.984	
	30	0.013 ± 0.001 ^C^	0.38 ± 0.013 ^C^	0.035	0.997	
	Xylanase SRXL1					
DK-2061	No inhibitor	4.34 ± 0.101 ^A^	1.090 ± 0.190 ^A^	3.984	0.999	Uncompetitive
	2.5	2.97 ± 0.109 ^B^	0.771 ± 0.040 ^B^	3.857	0.944	
	5.0	2.00 ± 0.049 ^C^	0.635 ± 0.039 ^C^	3.153	0.970	
BOGUI	No inhibitor	4.343 ± 0.101 ^A^	1.090 ± 0.190 ^A^	3.984	0.999	Uncompetitive
	2.5	3.368 ± 0.192 ^B^	0.944 ± 0.048 ^B^	3.567	0.996	
	5.0	1.857 ± 0.060 ^C^	0.056 ± 0.059 ^C^	3.281	0.988	

Capital letters show the results of statistical analyses, with identical letters between treatments indicating no statistically significant differences (*p* ≤ 0.05). The results of the statistical analysis are available in Appendix A. The determinations were performed in triplicate.

## Data Availability

Commercial hybrid corn seeds were used in this study. The white DK-2061variety (DeKalb^®^, Bayer, IL, USA) was purchased from a commercial seed company in Hidalgo, Mexico. The purple variety BOGUI (Biosuva, Mixquiahuala, Hidalgo, Mexico) was kindly donated by Engineer Jaime Ortega Bernal from the Mezquital Valley Centre for Innovation and Technological Development, Ministry of Agriculture and Rural Development, Hidalgo. Further inquiries can be directed to the corresponding author. The original contributions presented in this study are included in the article/Appendix A. Further inquiries can be directed to the corresponding authors.

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
