# Peer review of "Enzymatic Inhibitors of Aspartyl Protease EAP1 and Xylanase SRXL1 from Sporisorium reilianum Isolated from Corn Seeds"

_ijms, 2025, doi:10.3390/ijms26209974_

Round 1
Reviewer 1 Report
Comments and Suggestions for Authors
The paper titled "Enzymatic inhibitor of aspartyl protease EAP1 and xylanase 2 SRXL1 from Sporisorium reilianum isolated from corn seeds" presents interesting research on enzymatic inhibitors isolated from the seeds of two corn varieties (white – DK-2061 and purple – Ndaji), which exhibit inhibitory activity against two key enzymes of the fungus Sporisorium reilianum – aspartyl protease EAP1 and xylanase SRXL1. Of great scientific value is the fact that the authors were the first to demonstrate the ability of modified starch (acetylated during the purification process) to inhibit enzymes derived from S. reilianum. Importantly, despite the lack of direct antifungal activity, the paper indicates the potential use of the tested compounds as elements of plant protection strategies that limit the enzymatic activity of the pathogen without the use of chemical fungicides. However, the work raises several questions and inaccuracies, the explanation of which would be useful in this work:
Abstract
Line 23-24 I have the impression that the authors don't understand the characteristics of noncompetitive inhibition. Let me just remind you that in the case of noncompetitive inhibition, only one of the two values (Km or Vmax) changes.
Introduction section
Line 32-33 "….. but also as a raw material in various industries that create numerous products." This sentence adds no information. Please provide at least two examples of these "numerous products."
Results section
Line 133-134 In what units did the authors express the values for both DK-2061 hybrid and Ndaji?
Line 286-289. In this section of the work, the authors clearly indicate a noncompetitive type of inhibition, in which two kinetic values (Km and Vmax) change simultaneously. How is this possible? I have the impression that the authors lack sufficient knowledge of the kinetics and types of enzymatic inhibition. Let me remind you again that in the case of non-competitive inhibition, only one of the two values (Km or Vmax) changes.
Discussion section
Line 309-312 of Table 5. In this table (as in Table 6), the authors provide a completely different type of inhibition (Uncompetitive) than the previous one (non-competitive). Therefore, please choose one of the correct options.
Shouldn't the data presented in Tables 5 and 6 and Figures 5 and 6 be placed in the "Results section"?
Author Response
Response to reviewer
In response to the comments and suggestions regarding the manuscript entitled “Enzymatic inhibitor of aspartyl protease EAP1 and xylanase SRXL1 from Sporisorium reilianum isolated from corn seeds” submitted to International Journal of Molecular Sciences, the following aspects are discussed:
We apologize for the delay in replying to your comments, but we had to conduct additional experiments to verify our findings. These are shown on lines 304 to 333 and 696 to 722. It’s important to note that the document was reviewed for English copyediting by MDPI’s service.
Reviewer 1
Initial comment:
The paper titled "Enzymatic inhibitor of aspartyl protease EAP1 and xylanase 2 SRXL1 from Sporisorium reilianum isolated from corn seeds" presents interesting research on enzymatic inhibitors isolated from the seeds of two corn varieties (white – DK-2061 and purple – Ndaji), which exhibit inhibitory activity against two key enzymes of the fungus Sporisorium reilianum – aspartyl protease EAP1 and xylanase SRXL1. Of great scientific value is the fact that the authors were the first to demonstrate the ability of modified starch (acetylated during the purification process) to inhibit enzymes derived from S. reilianum. Importantly, despite the lack of direct antifungal activity, the paper indicates the potential use of the tested compounds as elements of plant protection strategies that limit the enzymatic activity of the pathogen without the use of chemical fungicides. However, the work raises several questions and inaccuracies, the explanation of which would be useful in this work:
Comment 1:
Abstract
Line 23-24 I have the impression that the authors don't understand the characteristics of noncompetitive inhibition. Let me just remind you that in the case of noncompetitive inhibition, only one of the two values (Km or Vmax) changes.
Response:
We have thoroughly reviewed the concepts of enzyme inhibition. In this case, uncompetitive inhibition occurs when the inhibitor binds to the enzyme-substrate complex rather than to the free enzyme, causing decreases in both Vmax and Km. Conversely, in non-competitive inhibition, the inhibitor can bind to either the free enzyme or the enzyme-substrate complex. In this case, Vmax decreases, while Km remains constant. This discrepancy may have arisen because we made an error on line 358, where we stated that the inhibition is non-competitive; in fact, it is uncompetitive. The correction has been made. The following reference provides information on this.
Patadiya, N.; Panchal, N.; Vaghela, V. A review on enzyme inhibitors. Int. Res. J. Pharm 2021, 12, 60-66. http://dx.doi.org/10.7897/2230-8407.1206145
Comment 2:
Introduction section
Line 32-33 "….. but also as a raw material in various industries that create numerous products." This sentence adds no information. Please provide at least two examples of these "numerous products."
Response:
In this regard, we attach some examples. Line 45
Comment 3:
Results section
Line 133-134 In what units did the authors express the values for both DK-2061 hybrid and Ndaji?
Response:
This paragraph explains which fractions from the ion-exchange chromatography purification process showed 100% inhibitory activity, so units are not necessary. For clarity, the sentence has been revised and restructured (Lines 127-129).
Comment 4:
Line 286-289. In this section of the work, the authors clearly indicate a noncompetitive type of inhibition, in which two kinetic values (Km and Vmax) change simultaneously. How is this possible? I have the impression that the authors lack sufficient knowledge of the kinetics and types of enzymatic inhibition. Let me remind you again that in the case of non-competitive inhibition, only one of the two values (Km or Vmax) changes.
Response:
As mentioned in the response to comment 1, we have reviewed the concepts of uncompetitive and non-competitive inhibition, and we have already corrected an error.
Comment 5:
Discussion section
Line 309-312 of Table 5. In this table (as in Table 6), the authors provide a completely different type of inhibition (Uncompetitive) than the previous one (non-competitive). Therefore, please choose one of the correct options.
Response:
We greatly appreciate your feedback, as we had this error, which, as already mentioned, has been corrected.
Comment 5:
Shouldn't the data presented in Tables 5 and 6 and Figures 5 and 6 be placed in the "Results section"?
Response:
That's correct, we modified the document so that they remain in the results section.
Corrections are highlighted in yellow in the manuscript.
Your comments and suggestions will contribute significantly to improving this manuscript.
Thank you for your observations,
Dr. Yuridia Mercado-Flores
Reviewer 2 Report
Comments and Suggestions for Authors
- Delete the dot at the end of the title.
- The abstract begins abruptly by mentioning the extraction of enzyme inhibitors, without providing any background or justification for this study. In my opinion, I suggest writing a brief introductory sentence highlighting the importance of controlling Sporisorium reilianum or the role of enzyme inhibition in plant protection.
- Line 19; "The identified inhibitors were starches" is ambiguous. Therefore, it would be helpful to clarify whether this result was unexpected or whether starches were the target compounds from the beginning.
- The sentence "During purification, the starches underwent acetylation" is unclear and confusing. Please rephrase it for clarity.
- Please note that lines 36-42 contain important information but are missing references.
- Please avoid single-sentence paragraphs all over your MS.
- In the Materials and Methods section (line 435), the authors stated that Sporisorium reilianum was isolated from corn crops and obtained from a colleague. However, they did not provide any description of the corn plants used for the isolation, such as the variety, growth stage, or health status, which is important for reproducibility and understanding the context of the pathogen source.
- Line 462, please italicize the scientific name Sporisorium reilianum in this line. Also, ensure that all scientific names throughout the manuscript are consistently italicized.
- Line 465; Clarification of the term "pre-inocula at were prepared": There seems to be a typographical error in the phrase “Two pre-inocula were prepared.” The sentence should be revised for clarity.
- While it is stated that the crude enzyme extract (CEE) was stored at -4°C, it would be helpful to clarify the maximum storage duration before use and whether enzyme stability was checked over time.
- Line 570, “the reddish color” could be made more precise (“pink endpoint” or “phenolphthalein endpoint”).
- Please report the number of replicates (n) for each assay in the figure/table legends.
- The manuscript mentions ANOVA and Tukey-Kramer post hoc tests, but relevant F-values, degrees of freedom, and exact p-values are nowhere reported in your text. This cannot be accepted.
- The manuscript refers to the inhibition as both “uncompetitive” and “non-competitive” in different sections (lines 287–289 and in the abstract). These are distinct inhibition types. Based on the decrease in both Km and Vmax, uncompetitive inhibition appears to be correct; however, this must be consistently clarified across the abstract, results, discussion, and tables.
- The presentation of the results (Figure 1) related to enzyme inhibition is minimal. It is advisable to include the actual inhibition percentages for each treatment in the text. Additionally, the figure caption should clearly state whether the error bars represent standard deviation (SD) or standard error (SE), and the number of replicates (n) should be indicated.
- Table 1 lacks essential details regarding the data presented. It is not clear whether the values are means of a certain number of replicates (n), nor is it specified whether the accompanying errors represent standard deviation (SD) or standard error of the mean (SE). This information should be explicitly stated in the table footnotes to ensure proper interpretation of the results.
- In Tables 5 and 6, the reductions in Vmax and Km suggest uncompetitive inhibition. While the Lineweaver–Burk plots support this, the authors should provide regression coefficients (R² values) or error estimates for kinetic models to support the claims quantitatively.
- Consider simplifying long and overly complex sentences, especially in the Introduction and Discussion sections.
Author Response
Response to reviewer
In response to the comments and suggestions regarding the manuscript entitled “Enzymatic inhibitor of aspartyl protease EAP1 and xylanase SRXL1 from Sporisorium reilianum isolated from corn seeds” submitted to International Journal of Molecular Sciences, the following aspects are discussed:
We apologize for the delay in replying to your comments, but we had to conduct additional experiments to verify our findings. These are shown on lines 304 to 333 and 696 to 722. It’s important to note that the document was reviewed for English copyediting by MDPI’s service.
Reviewer 2
Comment 1:
Delete the dot at the end of the title.
Response:
The correction was made
Comment 2:
The abstract begins abruptly by mentioning the extraction of enzyme inhibitors, without providing any background or justification for this study. In my opinion, I suggest writing a brief introductory sentence highlighting the importance of controlling Sporisorium reilianum or the role of enzyme inhibition in plant protection.
Response:
The abstract was supplemented with the requested information. (Lines 16-22)
Comment 3:
Line 19; "The identified inhibitors were starches" is ambiguous. Therefore, it would be helpful to clarify whether this result was unexpected or whether starches were the target compounds from the beginning.
Response:
The sentence was completed (Lines 25-26)
Comment 4:
The sentence "During purification, the starches underwent acetylation" is unclear and confusing. Please rephrase it for clarity.
Response:
The sentence was modified for clarity (Lines 29-30)
Comment 5:
Please note that lines 36-42 contain important information but are missing references.
Response:
The references corresponding to the information in said paragraph are found on line 58.
Comment 6:
Please avoid single-sentence paragraphs all over your MS.
Response:
The entire document was reviewed and corrections were made.
Comment 7:
In the Materials and Methods section (line 435), the authors stated that Sporisorium reilianum was isolated from corn crops and obtained from a colleague. However, they did not provide any description of the corn plants used for the isolation, such as the variety, growth stage, or health status, which is important for reproducibility and understanding the context of the pathogen source.
Response:
In the materials and methods, it is mentioned that the phytopathogen was isolated from the teliospores of diseased plants in a state of physiological maturity (Lines 538-540)
Comment 8:
Line 462, please italicize the scientific name Sporisorium reilianum in this line. Also, ensure that all scientific names throughout the manuscript are consistently italicized.
Response:
The corresponding correction was made. (Lines 567-568)
Comment 9:
Line 465; Clarification of the term "pre-inocula at were prepared": There seems to be a typographical error in the phrase “Two pre-inocula were prepared.” The sentence should be revised for clarity.
Response:
The correction was made. (Line 570)
Comment 10:
While it is stated that the crude enzyme extract (CEE) was stored at -4°C, it would be helpful to clarify the maximum storage duration before use and whether enzyme stability was checked over time.
Response:
The document stated that storage was intended for a duration of 8 days (Line 574). Throughout this period, the activity of the crude enzyme extracts remained at 100%. Enzyme activity was not assessed over time, as this was not the objective of the study.
Comment 11:
Line 570, “the reddish color” could be made more precise (“pink endpoint” or “phenolphthalein endpoint”).
Response:
It was specified more precisely, for greater clarity (Line 674-675)
Comment 12:
Please report the number of replicates (n) for each assay in the figure/table legends.
Response:
The number of replicates was specified in the legends of the tables and figures.
Comment 13:
The manuscript mentions ANOVA and Tukey-Kramer post hoc tests, but relevant F-values, degrees of freedom, and exact p-values are nowhere reported in your text. This cannot be accepted.
Response:
In results where statistical analysis was performed, the p-value is indicated in the table footnote. To enhance clarity, the results of the statistical analyses are provided as supplementary material.
Comment 14:
The manuscript refers to the inhibition as both “uncompetitive” and “non-competitive” in different sections (lines 287–289 and in the abstract). These are distinct inhibition types. Based on the decrease in both Km and Vmax, uncompetitive inhibition appears to be correct; however, this must be consistently clarified across the abstract, results, discussion, and tables.
Response:
Thank you for your feedback. This correction was made throughout the document. The correct term is uncompetitive.
Comment 15:
The presentation of the results (Figure 1) related to enzyme inhibition is minimal. It is advisable to include the actual inhibition percentages for each treatment in the text. Additionally, the figure caption should clearly state whether the error bars represent standard deviation (SD) or standard error (SE), and the number of replicates (n) should be indicated.
Response:
The inhibition depicted in Figure 1 is significant; the percentage of inhibition is graphed, showing 100% for the xylanase SRXL1, while the aspartyl protease EAP1 exhibited over 70% inhibition. These values are included in the text (lines 119-123). Additionally, the figure caption notes that the error bars represent the standard deviation (line 164).
Comment 16:
Table 1 lacks essential details regarding the data presented. It is not clear whether the values are means of a certain number of replicates (n), nor is it specified whether the accompanying errors represent standard deviation (SD) or standard error of the mean (SE). This information should be explicitly stated in the table footnotes to ensure proper interpretation of the results.
Response:
The footnote to Table 1 specifies that the values ​​represent the average of three replicates along with their standard deviation (Line 171).
Comment 17:
In Tables 5 and 6, the reductions in Vmax and Km suggest uncompetitive inhibition. While the Lineweaver–Burk plots support this, the authors should provide regression coefficients (R² values) or error estimates for kinetic models to support the claims quantitatively.
Response:
The R2 values ​​were incorporated into Tables 5 and 6.
Comment 18:
Consider simplifying long and overly complex sentences, especially in the Introduction and Discussion sections.
Response:
The document was reviewed for style correction.
Corrections are highlighted in turquoise blue in the manuscript.
Your comments and suggestions will contribute significantly to improving this manuscript.
Thank you for your observations,
Dr. Yuridia Mercado-Flores
Reviewer 3 Report
Comments and Suggestions for Authors
The article describes an inhibitor of polysaccharide activity, but the literature review contains references only to works on peptides. It is necessary to add references to the literature on similar substances. Line 105 - Starch does not dissolve in cold water, but in hot water it swells and turns into paste. Line 443 - You also did not add acids to improve starch solubility. So how could the "starch" in the aqueous extract inhibit something? Could it have been other substances? How do you confirm that this activity was due to starches? The fact that you analyzed the extracts using ion exchange chromatography and determined the starch peaks does not mean that they were the cause of the inhibition. Starches may simply be pollutants in the purified extract. It seems to me that it is necessary to put some control. Heat the extracts to 80 degrees and look at their inhibitory activity to exclude the effect of inhibitor peptides in the extract. It seems to me that the theory of indirect inhibition is not valid due to the lack of control over the presence of peptide inhibitors in extracts. And there are no attempts to inactivate them. Scanning electron microscopy is used simply for morphological description of starch grains. Why not do the xylanase treatment and describe the resulting effects? They should correlate with your findings. And in this form, I don't understand why SEM is here. EDS is not applicable here at all, I do not understand why it is in the study. Two halves of one split starch grain can give different percentages of the elements. What qualitative important information did the EDS analysis provide for the study? The answer is no. But the lack of research on the presence of peptide inhibitors in the extract casts doubt on the entire study design and conclusions. The summary of the study looks extremely dubious, primarily due to the lack of controls.
Author Response
Reviewer 3
Comment 1:
The article describes an inhibitor of polysaccharide activity, but the literature review contains references only to works on peptides. It is necessary to add references to the literature on similar substances. Line 105 - Starch does not dissolve in cold water, but in hot water it swells and turns into paste. Line 443 - You also did not add acids to improve starch solubility. So how could the "starch" in the aqueous extract inhibit something? Could it have been other substances? How do you confirm that this activity was due to starches? The fact that you analyzed the extracts using ion exchange chromatography and determined the starch peaks does not mean that they were the cause of the inhibition. Starches may simply be pollutants in the purified extract. It seems to me that it is necessary to put some control. Heat the extracts to 80 degrees and look at their inhibitory activity to exclude the effect of inhibitor peptides in the extract. It seems to me that the theory of indirect inhibition is not valid due to the lack of control over the presence of peptide inhibitors in extracts. And there are no attempts to inactivate them. Scanning electron microscopy is used simply for morphological description of starch grains. Why not do the xylanase treatment and describe the resulting effects? They should correlate with your findings. And in this form, I don't understand why SEM is here. EDS is not applicable here at all, I do not understand why it is in the study. Two halves of one split starch grain can give different percentages of the elements. What qualitative important information did the EDS analysis provide for the study? The answer is no. But the lack of research on the presence of peptide inhibitors in the extract casts doubt on the entire study design and conclusions. The summary of the study looks extremely dubious, primarily due to the lack of controls.
Response:
So far, reports have only documented enzyme inhibitors of proteases and xylanases derived from plants and seeds of a peptide nature; this work is the first to report starch as an enzyme inhibitor.
Starch is insoluble in cold water but may be partially soluble in room temperature water, depending on the ratio of amylose and amylopectin.
https://doi.org/10.1016/B978-0-12-398358-9.00023-9
An additional experiment was conducted to verify that enzyme inhibition is caused by starches. Commercial α-amylase enzyme from Bacillus amyloliquefaciens was used, with flours from seeds of the two hybrids and purified inhibitors as substrates. Results showed that after the enzymatic reaction, all samples lost their activity. The procedure for this experiment, along with the controls, is detailed in the materials and methods section, and the findings are reported in the corresponding section. Additionally, this result was included in the abstract (Lines 31-32, 304-332, 696-722).
The aqueous extracts and purified inhibitors were subjected to a heating process for 1 hour to subsequently measure their inhibitory effect. However, they underwent a gelation process, which is common for starch (https://doi.org/10.3390/polym16050597), so the sample could not be used to perform the inhibition tests.
We consider that the tests performed with α-amylase support the conclusion that starches are the substances responsible for inhibiting the enzymes studied.
SEM and EDS studies were used to characterize the inhibitors under investigation. While EDS results can vary depending on the area analyzed, the data presented are from three replicates.
Corrections are highlighted in grey in the manuscript.
Your comments and suggestions will contribute significantly to improving this manuscript.
Thank you for your observations,
Dr. Yuridia Mercado-Flores
Reviewer 4 Report
Comments and Suggestions for Authors
Comments to the author (IJMS-3806047)
The manuscript titled “Enzymatic inhibitor of aspartyl protease EAP1 and xylanase SRXL1 from Sporisorium reilianum isolated from corn seeds” by Yusiri Velázquez-Juárez is an interesting study which investigates enzyme inhibitors from two corn hybrids (DK-2061 and Ndaji) that inhibit aspartyl protease EAP1 and xylanase SRXL1 enzymes produced by Sporisorium reilianum, the fungus causing corn ear smut. The inhibitors, identified as starches, exhibit uncompetitive inhibition, offering a potential alternative to chemical fungicides for disease control. Overall, the study is presenting novelty and the manuscript is well structured; presenting the merit of literature and is in accordance with the Journal’s scope. However, some short comings in text are needed to be fixed for the improvement of manuscript.
Remove “.” From title
Please add some numeric values in abstract to support the results (where applicable)
In line 19: “The inhibitors identified were starches.” Needs rephrasing as it is too short.
In line 22: “but retained their inhibitory activity” Please clarify the extent of activity retention (e.g., partial/complete).
In line 23-24: “evidenced by a decrease in the Km and Vmax values.” Please specify here that it was determined by Lineweaver-Burk analysis for accuracy.
In line 40: “optimal temperature and humidity condition” Should be plural: “conditions.”
In line 99: “to evaluate their effects on the enzymatic activities...” please rephrase “to evaluate their inhibitory effects...” for specificity and also add the variety names.
In Figure 1: The error bars are not clearly visible in parts A and B; ensure they are readable and state sample size (n) in legend and also, the vertical axis caption is missing.
In table 1: Please clarify how “100% inhibition” was determined (e.g., from triplicate average?); include statistical comparison if applicable.
In line 152-154: “The band around 990 cm-1 shifted...” please confirm if this shift indicates successful acetylation; interpretation is implied but not explicitly stated.
In line 158: “acetylation ester of the inhibitors...” needed to be rephrase “acetyl ester functional group,” which is more chemically accurate.
In line 168: remove the abbreviation from heading and cross check in all subheadings. Also, please mention the full as in first mention and later use the abbreviation (where needed).
In line 169: do not start the sentence with abbreviation.
Line 173: “Starch granules were observed in the aqueous extracts…” please add confirmation whether these were intact or swollen this affects interpretation of inhibitor accessibility.
In line 180: “crystals of the salts from the regulators…” please consider specifying the chemical composition (e.g., NaCl?) based on EDS data.
Figure 2: Is it possible to add baseline FT-IR for commercial starch control for easier comparison?
In figure 3: Include scale bars on all SEM micrographs for consistency and better interpretation.
In figure 4: Same comment as above: add scale bars, and better contrast to show surface porosity.
In line 244: In EDS analysis, please include standard deviation in elemental composition table (Table 2) or mention if values are averages.
In line 263-267: in “Degree of acetylation” please clearly state what the acceptable or functional threshold is (e.g., compare to literature) to interpret significance.
In line 271: “Km and Vmax…” include R2 values or residuals from the regression fits if available.
Figures 5 and 6 present graphs. The legends of figures should clearly mention which lines are for control vs. treatment. Also, in the axis labels there is a formatting/linguistic error.
In line 292-294: “...no antifungal activity.” As this is the key finding of the study I would suggest to elaborate and discuss the possible reasons (e.g., size, inability to penetrate fungal cell wall).
In line 366-371: “...present uncompetitive inhibition…” There is an inconsistency: Abstract says uncompetitive; results/discussion sometimes say noncompetitive. Resolve this contradiction.
In line 423: “...can be observed individually or in groups.” citation needed to support this claim.
In line 430-432: “...by blocking the extracellular activities…” My personal suggestion is to add a short qualifier that these findings are in vitro and may not fully translate to in planta control.
In line 446: “...left to dry for 96 hours in a dryer.” Please specify the dryer temperature and humidity, as drying conditions can affect starch structure.
In line 522: “...filtrate was passed through a Q Sepharose...” need to column volume and binding/elution buffer details for reproducibility.
In line 559: “...I2/KI solution...” what was the final concentrations or volume ratio used in the test for replicability?
Some subheadings are too long and not scientifically attractive for reader, for example, in line 563: change the subheading to “Acetylation degree of inhibitors” and same for results sections.
Similarly, 269 should be “Kinetic Effects on EAP1 and SRXL1” and in line 103: should be “Inhibitory Activity of Corn Seed Extracts”
In line 594: “...Km and Vmax of the enzymes under study…” please clarify which figure or table the reader should refer to for results.
In line 623: “...examined.” How the growth was quantified i.e colony diameter, visual scoring or what?
In line 627: “Statistical analyses were conducted using...” Whether the data were checked for normality and variance homogeneity before ANOVA.
Some references are too old, for example, reference 3, 8, 35, 37, 54, 60, 61, 69. I would suggest removing these references and replacing with the updated ones. If these are the established protocols, then try to replace with the latest and updated ones.
Comments on the Quality of English LanguagePlease refer to detailed comments
Author Response
Response to reviewer
In response to the comments and suggestions regarding the manuscript entitled “Enzymatic inhibitor of aspartyl protease EAP1 and xylanase SRXL1 from Sporisorium reilianum isolated from corn seeds” submitted to International Journal of Molecular Sciences, the following aspects are discussed:
We apologize for the delay in replying to your comments, but we had to conduct additional experiments to verify our findings. These are shown on lines 304 to 333 and 696 to 722. It’s important to note that the document was reviewed for English copyediting by MDPI’s service.
Reviewer 4
The manuscript titled “Enzymatic inhibitor of aspartyl protease EAP1 and xylanase SRXL1 from Sporisorium reilianum isolated from corn seeds” by Yusiri Velázquez-Juárez is an interesting study which investigates enzyme inhibitors from two corn hybrids (DK-2061 and Ndaji) that inhibit aspartyl protease EAP1 and xylanase SRXL1 enzymes produced by Sporisorium reilianum, the fungus causing corn ear smut. The inhibitors, identified as starches, exhibit uncompetitive inhibition, offering a potential alternative to chemical fungicides for disease control. Overall, the study is presenting novelty and the manuscript is well structured; presenting the merit of literature and is in accordance with the Journal’s scope. However, some short comings in text are needed to be fixed for the improvement of manuscript.
Comment 1:
Remove “.” From title
Response:
The observation was attended to (Line 3)
Comment 2:
Please add some numeric values in abstract to support the results (where applicable)
Response:
Numerical values supporting the results were included in the abstract (Lines 25-26 and 29).
Comment 3:
In line 19: “The inhibitors identified were starches.” Needs rephrasing as it is too short.
Response:
The sentence was completed in abstract (lines 25-26).
Comment 4:
In line 22: “but retained their inhibitory activity” Please clarify the extent of activity retention (e.g., partial/complete).
Response:
The abstract specified that the inhibitory activity was complete (Lines 30-31).
Comment 5:
In line 23-24: “evidenced by a decrease in the Km and Vmax values.” Please specify here that it was determined by Lineweaver-Burk analysis for accuracy.
Response:
The abstract specifies that the values ​​of Km and Vax were determined using the Lineweaver-Burk equation (Line 34).
Comment 6:
In line 40: “optimal temperature and humidity condition” Should be plural: “conditions.”
Response:
The change was made (Line 52)
Comment 7:
In line 99: “to evaluate their effects on the enzymatic activities...” please rephrase “to evaluate their inhibitory effects...” for specificity and also add the variety names.
Response:
Lines 111 and 112 specify the types of corn used.
Comment 8:
In Figure 1: The error bars are not clearly visible in parts A and B; ensure they are readable and state sample size (n) in legend and also, the vertical axis caption is missing.
Response:
An attempt was made to improve the image; however, the error bars show low values. The figure legend specifies that the experiments were performed in triplicate (Lines 163-164). The vertical axis indicates the percentage of inhibition, as specified in the figure.
Comment 9:
In table 1: Please clarify how “100% inhibition” was determined (e.g., from triplicate average?); include statistical comparison if applicable.
Response:
The footnote to Table 1 specifies that the determinations were done in triplicate (Line 171). In this case, we did not consider performing statistical analysis because the samples are fractions from the chromatography processes.
Comment 10:
In line 152-154: “The band around 990 cm-1 shifted...” please confirm if this shift indicates successful acetylation; interpretation is implied but not explicitly stated.
Response:
Lines 191-192 specify that this shift indicates a structural change, likely caused by the acetylation process.
Comment 11:
In line 158: “acetylation ester of the inhibitors...” needed to be rephrase “acetyl ester functional group,” which is more chemically accurate.
Response:
The correction was made (Line 197).
Comment 12:
In line 168: remove the abbreviation from heading and cross check in all subheadings. Also, please mention the full as in first mention and later use the abbreviation (where needed).
Response:
The changes were made according to the suggestion.
Comment 13:
In line 169: do not start the sentence with abbreviation.
Response:
The change was made according to the suggestion (Line 202).
Comment 14:
Line 173: “Starch granules were observed in the aqueous extracts…” please add confirmation whether these were intact or swollen this affects interpretation of inhibitor accessibility.
Response:
Line 206 states that the starch granules remained intact. It is worth noting that the aqueous extracts were lyophilized for SEM analysis, as specified in lines 655-656.
Comment 15:
In line 180: “crystals of the salts from the regulators…” please consider specifying the chemical composition (e.g., NaCl?) based on EDS data.
Response:
In line 212, the chemical composition of the salts was specified, which in this case was NaCl.
Comment 16:
Figure 2: Is it possible to add baseline FT-IR for commercial starch control for easier comparison?
Response:
In Figure 2, the yellow line shows commercial corn starch, as indicated in the legend. The use of commercial starch is described in the Materials and Methods section for comparison with the samples (Line 649).
Comments 17 and 18:
In figure 3: Include scale bars on all SEM micrographs for consistency and better interpretation.
In figure 4: Same comment as above: add scale bars, and better contrast to show surface porosity.
Response:
Figures 3 and 4 were adjusted to improve the visibility of the scale bars.
Comment 19:
In line 244: In EDS analysis, please include standard deviation in elemental composition table (Table 2) or mention if values are averages.
Response:
In Table 2, the values are expressed with their standard deviation and are the averages of the measurements; the latter is specified in the table footnote (Line 286).
Comment 20:
In line 263-267: in “Degree of acetylation” please clearly state what the acceptable or functional threshold is (e.g., compare to literature) to interpret significance.
Response:
The discussion mentions the comparison of the results obtained with respect to the literature (Lines 509-519).
Comment 21:
In line 271: “Km and Vmax…” include R2 values or residuals from the regression fits if available.
Response:
The R2 values ​​were included in Tables 7 and 8.
Comment 22:
Figures 5 and 6 present graphs. The legends of figures should clearly mention which lines are for control vs. treatment. Also, in the axis labels there is a formatting/linguistic error.
Response:
The figure indicates what each line represents. The axis labels have been reviewed and adjusted.
Comment 23:
In line 292-294: “...no antifungal activity.” As this is the key finding of the study I would suggest to elaborate and discuss the possible reasons (e.g., size, inability to penetrate fungal cell wall).
Response:
In lines 527-430, the effect of starch on the development of S. reilianum is discussed.
Comment 24:
In line 366-371: “...present uncompetitive inhibition…” There is an inconsistency: Abstract says uncompetitive; results/discussion sometimes say noncompetitive. Resolve this contradiction.
Response:
In this case, we had a translation error where we stated that the inhibition is non-competitive, when it's uncompetitive. The correction was applied throughout the document.
Comment 25:
In line 423: “...can be observed individually or in groups.” citation needed to support this claim.
Response:
In this case, reference 62 was used (line 523).
Comment 26:
In line 430-432: “...by blocking the extracellular activities…” My personal suggestion is to add a short qualifier that these findings are in vitro and may not fully translate to in planta control.
Response:
The suggestion was considered, and the wording in lines 533-535 was revised.
Comment 27:
In line 446: “...left to dry for 96 hours in a dryer.” Please specify the dryer temperature and humidity, as drying conditions can affect starch structure.
Response:
Lines 552-553 describe the temperature and humidity conditions to which the seeds were subjected.
Comment 28:
In line 522: “...filtrate was passed through a Q Sepharose...” need to column volume and binding/elution buffer details for reproducibility.
Response:
Line 628 indicates the column volume. Lines 630-632 detail the conditions for setting up the elution gradient.
Comment 29:
In line 559: “...I2/KI solution...” what was the final concentrations or volume ratio used in the test for replicability?
Response:
Lines 666-667 specify the final concentrations of the I2/KI solution components in the reaction mixture.
Comment 30:
Some subheadings are too long and not scientifically attractive for reader, for example, in line 563: change the subheading to “Acetylation degree of inhibitors” and same for results sections.
Response:
We appreciate your comment, but we believe the subheadings give a detailed description of the activities performed and the results shown.
Comment 31:
Similarly, 269 should be “Kinetic Effects on EAP1 and SRXL1” and in line 103: should be “Inhibitory Activity of Corn Seed Extracts”
Response:
We also appreciate your comment, but we believe the subheadings offer a detailed description of the activities conducted and the results shown.
Comment 32:
In line 594: “...Km and Vmax of the enzymes under study…” please clarify which figure or table the reader should refer to for results.
Response:
The paragraph on lines 726-730 was modified to specify the concentrations of the purified inhibitors used to measure their effect on the kinetic parameters of the Eap1 and SRXLI enzymes.
Comment 33:
In line 623: “...examined.” How the growth was quantified i.e colony diameter, visual scoring or what?
Response:
In lines 365-366 of the results, it is noted that the yeast-like growth was observed across the entire surface of the medium on the plate; similarly, in the materials and methods section (lines 761-762), it is specified that the reading was done visually.
Comment 35:
In line 627: “Statistical analyses were conducted using...” Whether the data were checked for normality and variance homogeneity before ANOVA.
Response:
Lines 721-723 state that the data were tested for normal distribution and homoscedasticity using the Shapiro–Wilk or Kolmogorov-Smirnov tests and Levene's test, respectively. This information was also included in the supplementary material.
Comment 36:
Some references are too old, for example, reference 3, 8, 35, 37, 54, 60, 61, 69. I would suggest removing these references and replacing with the updated ones. If these are the established protocols, then try to replace with the latest and updated ones.
Response:
We believe it is important to cite the authors who initially reported the works we used to support this manuscript.
Corrections are highlighted in green in the manuscript.
Your comments and suggestions will contribute significantly to improving this manuscript.
Thank you for your observations,
Dr. Yuridia Mercado-Flores
Round 2
Reviewer 2 Report
Comments and Suggestions for Authors
The authors have performed the required revisions satisfactorily.
Author Response
In response to the comments and suggestions regarding the manuscript entitled “Enzymatic inhibitor of aspartyl protease EAP1 and xylanase SRXL1 from Sporisorium reilianum isolated from corn seeds” submitted to International Journal of Molecular Sciences, the following aspects are discussed:
Reviewer 2
Comment 1:
The authors have performed the required revisions satisfactorily.
Response:
Thank you very much for your review. Your comments and suggestions will contribute significantly to improving this manuscript.
Thank you for your observations,
Dr. Yuridia Mercado-Flores
Reviewer 3 Report
Comments and Suggestions for Authors
I read the author's answer and I'm at a loss. The authors took a calcium-dependent enzyme from another family and conducted an experiment without calcium ions in the medium. And then they show the inhibition of the enzyme on the substrate that this enzyme IS SUPPOSED to cleave. And stay active at the same time. It is possible, of course, to overload the hydrolysis with a large amount of substrate and obtain inhibition due to a non-specific interaction. This is not inhibition, it is a misrepresentation of the experience. Something went wrong with the inhibition of enzymes from extracts by temperature, which is extremely doubtful. I didn't get an answer to the question of why SEM EDS is needed here at all, why are there any repetitions in the answer at all. The question was about something else and needs no explanation. Maybe I don't understand something and the authors are revolutionaries in their field. There is not a single article in the world literature on the inhibition of starch substrate by an enzyme aimed at its hydrolysis. And this is logical. Reject.
Author Response
In response to the comments and suggestions regarding the manuscript entitled “Enzymatic inhibitor of aspartyl protease EAP1 and xylanase SRXL1 from Sporisorium reilianum isolated from corn seeds” submitted to International Journal of Molecular Sciences, the following aspects are discussed:
Reviewer 3
Comment 1:
I read the author's answer and I'm at a loss. The authors took a calcium-dependent enzyme from another family and conducted an experiment without calcium ions in the medium. And then they show the inhibition of the enzyme on the substrate that this enzyme IS SUPPOSED to cleave. And stay active at the same time. It is possible, of course, to overload the hydrolysis with a large amount of substrate and obtain inhibition due to a non-specific interaction. This is not inhibition, it is a misrepresentation of the experience. Something went wrong with the inhibition of enzymes from extracts by temperature, which is extremely doubtful. I didn't get an answer to the question of why SEM EDS is needed here at all, why are there any repetitions in the answer at all. The question was about something else and needs no explanation. Maybe I don't understand something and the authors are revolutionaries in their field. There is not a single article in the world literature on the inhibition of starch substrate by an enzyme aimed at its hydrolysis. And this is logical. Reject.
Response:
Dear reviewer, we appreciate your comments, but before considering rejecting our work for publication, please consider the following points:
- First of all, we apologize for the delay in responding to your comments; we felt it was important to conduct an additional experiment to confirm our findings, which will be discussed later.
- Regarding the use of α-amylase, we apologize if our response to your comment was not clear enough. The purpose of using this enzyme was to confirm that the inhibitors extracted from corn were related to starches. Specifically, the plan was to use corn seed flours with inhibitory activity and the purified inhibitors as substrates for α-amylase. Our hypothesis was that if the enzyme degraded the starch in the flours or the purified inhibitors, then the inhibitors would lose their inhibitory activity. First, we confirmed that the enzyme was active against the flours and inhibitors; as a control, we used potato starch, a common substrate for detecting amylolytic activity. In Table 1, it is shown that, based on the definition of an enzymatic activity unit for α-amylase (Line 727-729), the enzyme used potato starch, flours extracted from the corn hybrids under study, and the purified inhibitors as substrates. Based on these results, a second experiment was performed. The results are presented in Table 2. The treatment involved incubating the enzyme with the flours and inhibitors as substrates (details are provided in section 4.13). Once the reaction was stopped, the resulting solution was used to measure its inhibitory effect on the aspartyl protease EAP1 and the xylanase SRXL1. It was observed that the flours and purified inhibitors lost their ability to inhibit the aforementioned enzymes after being degraded by α-amylase. Two controls were established: in the first, α-amylase was heat-inactivated before incubation with the flours and purified inhibitors. In the second control, distilled water was used instead of the enzyme. In both controls, the flours and purified inhibitors retained their inhibitory activity.
- You are right to mention how it is possible to determine the activity of α-amylase, a Ca-dependent enzyme, without adding this element to the reaction buffer. In this regard, we note that the enzyme used was the commercial α-amylase from Bacillus amyloliquefaciens (Sigma-Aldrich, A7595). The manufacturer's recommended protocol can be found at the following website: https://www.sigmaaldrich.com/MX/es/technical-documents/protocol/protein-biology/enzyme-activity-assays/enzymatic-assay-of-a-amylase, the recommended buffer is 20 mM Sodium Phosphate with 6.7 mM Sodium Chloride at pH 6.9, which does not contain Calcium ions; however, we observe enzymatic activity. It's worth noting that the commercially available α-amylase from amyloliquefaciens is provided in liquid form, which may imply that it contains calcium to maintain the enzyme's stability. However, the supplier's information does not specify the components of this product. In reviewing the literature, we found the following article: Gangadharan, D., Sivaramakrishnan, S., Nampoothiri, K. M., Sukumaran, R. K., & Pandey, A. 2008. Response surface methodology for the optimization of alpha amylase production by Bacillus amyloliquefaciens. Bioresource technology, 99(11), 4597–4602. https://doi.org/10.1016/j.biortech.2007.07.028, it is mentioned that the production of α-amylase in B. amyloliquefaciens depends on substrate concentrations (cheap agricultural waste), incubation period, and CaCl2 concentration. An optimized process resulted in crude extracellular enzyme extracts (CEE) with higher amylolytic activity, measured using starch as a substrate and a 0.1 M acetate buffer at pH 5.0 that lacked calcium ions. This suggests that the enzyme in the CEE is stable, and most likely, the calcium in the culture supernatant could contribute to this stability. In this case, the commercial α-amylase product, since its degree of purity is not specified, could be a CEE that may contain calcium, or the enzyme is contained in a buffer with the necessary components to maintain its stability.
- Regarding your suggestion to heat-inactivate potential peptic inhibitors that could be contaminating the purified inhibitors, we observed that when the samples were heated to 60, 80, and 100 °C for 1 h, a gel formed, making it difficult to take samples for the enzyme inhibition assay. The literature indicates that starch granules swell when heated above 50°C, thereby increases their viscosity (Zarski, A., Kapusniak, K., Ptak, S., Rudlicka, M., Coseri, S., & Kapusniak, J. 2024. Functionalization Methods of Starch and Its Derivatives: From Old Limitations to New Possibilities. Polymers, 16, 597). https://doi.org/10.3390/polym16050597).
- We apologize for not explaining our ideas clearly regarding the use of SEM and EDS. After obtaining the fractions with the purified inhibitors and conducting the FTIR analysis, we doubted whether the compounds responsible for the inhibition were starches. Therefore, we decided to perform an EDS analysis to search for the presence of nitrogen, which could indicate that the sample contained peptides. Since this analysis is combined with SEM, we observed the presence of starch granules and the absence of nitrogen in the sample, which we consider an important result.
- So far, there are no reports indicating that starch inhibits the aspartyl protease EAP1 and xylanase SRXL1 enzymes of reilianum; our study could be the first to document this.
- Regarding the use of α-amylase, as previously discussed, the flours and purified inhibitors were not used as inhibitors of this enzyme but rather as substrates. In this case, the purified inhibitors lost their inhibitory activity on the reilianum aspartyl protease EAP1 and xylanase SRXL1 enzymes after treatment with α-amylase. The inclusion of this experiment was motivated by your observations, for which we are grateful.
Thank you for taking the time to review our work.
Your comments and suggestions will contribute significantly to improving this manuscript.
Thank you for your observations,
Dr. Yuridia Mercado-Flores
Reviewer 4 Report
Comments and Suggestions for Authors
Comments to the author
The authors have adequately addressed all comments but still needed some minor improvements.
There are some grammatical mistakes and I would suggest to carefully revise it before final acceptance by IJMS.
I have following suggestions/corrections needed to be considered.
Line 24: I think this word is “with” not whit
Capitalized the first letter of each word in Key word
Line 49-57: Add references
Line 73: “managing phytosanitary problems” this is redundant wrt to present study.
Line 114-115: devastating fungal pathogen.
Fig 1: add horizontal caption
In table 1, 2, 4: Statistical alphabets letters for significant differences are missing.
Reduce the number of tables up to 4, it would be sufficient and shift some tables to supplementary data section.
I am confused, sometimes probably in the introduction section the authors have mentioned (in line 70-71: Bacillus velezensis as biocontrol agent) but in M&M section, the authors have stated the use of Bacillus amyloliquefaciens enzyme inhibitors. This makes a great confusion. Use of bacillus species as Biocontrol agents is somewhat different than use of enzyme inhibitors of these species. Particularly, in the present study, I think B. velezensis has nothing to do with but only B. amyloliquefaciens mainly its enzyme inhibitors. So, please add some background theme generally for all Bacillus and focusing on B. amyloliquefaciens.
Overall, the authors have used many enzyme inhibitors abbreviations, some they have used even at first mention. I would suggest to list these abbreviations at the end or use full name and then abbreviations.
Please add conclusions and future directions
Please refer to the comments
Author Response
In response to the comments and suggestions regarding the manuscript entitled “Enzymatic inhibitor of aspartyl protease EAP1 and xylanase SRXL1 from Sporisorium reilianum isolated from corn seeds” submitted to International Journal of Molecular Sciences, the following aspects are discussed:
Reviewer 4
The authors have adequately addressed all comments but still needed some minor improvements.
There are some grammatical mistakes and I would suggest to carefully revise it before final acceptance by IJMS.
I have following suggestions/corrections needed to be considered.
Comment 1:
Line 24: I think this word is “with” not whit
Response:
The change was made to Line 24.
Comment 2:
Capitalized the first letter of each word in Key word
Response:
The change was made. Lines 38-39.
Comment 3:
Line 49-57: Add references
Response:
References were added in the paragraph. Lines 49-59.
Comment 4:
Line 73: “managing phytosanitary problems” this is redundant wrt to present study.
Response:
The sentence was restructured. Line 74.
Comment 5:
Line 114-115: devastating fungal pathogen.
Response:
The correction was made. Lines 116-117
Comment 6:
Fig 1: add horizontal caption
Response:
The horizontal caption was added in Figure 1.
Comment 7:
In table 1, 2, 4: Statistical alphabets letters for significant differences are missing.
Response:
In Tables 1 and 2, no statistical analyses were conducted for the following reasons:
- Table 1 now Table S1. The results correspond to the inhibitory activity of different fractions obtained from chromatographic processes. The goal was to purify the inhibitors and identify the fractions with the highest inhibition percentage. In this type of procedure, we are not making a comparative analysis; these are simply results from an analytical technique.
- Table 2 now Table S2. The results in this table come from the EDS analysis system. The software selects three points on the sample and computes an average along with the standard deviation. It does not supply a database of individual values, so they cannot be used for statistical analysis. These data only show whether certain elements are present or absent in the sample; they were not utilized for comparative analysis that requires statistical information.
The results of the statistical analyses were included in Table 4, now Table 1. Lines 307-321. The statistical analysis results are provided in the supplementary material as Table S5. Lines 797-798.
Comment 8:
Reduce the number of tables up to 4, it would be sufficient and shift some tables to supplementary data section.
Response:
The number of tables was reduced to 4. Three of the original tables were included in the supplementary material (Lines 788-795). The original Tables 7 and 8 were combined to create Table 4 (Lines 385-391).
Comment 9:
I am confused, sometimes probably in the introduction section the authors have mentioned (in line 70-71: Bacillus velezensis as biocontrol agent) but in M&M section, the authors have stated the use of Bacillus amyloliquefaciens enzyme inhibitors. This makes a great confusion. Use of bacillus species as Biocontrol agents is somewhat different than use of enzyme inhibitors of these species. Particularly, in the present study, I think B. velezensis has nothing to do with but only B. amyloliquefaciens mainly its enzyme inhibitors. So, please add some background theme generally for all Bacillus and focusing on B. amyloliquefaciens.
Response:
We apologize for not communicating the idea clearly.
The purpose of using α-amylase from B. amyloliquefaciens was to verify that the inhibitors extracted from corn were related to starches. It was not used as an inhibitor. In this case, corn seed flours with inhibitory activity and purified inhibitors were used as substrates for α-amylase. Our hypothesis was that if the enzyme degraded the starch in the flours or the purified inhibitors, they would lose their inhibitory activity. First, we confirmed that the enzyme was active against the flours and inhibitors; as a control, we used potato starch, a common substrate for detecting amylolytic activity. In Table 1, it is shown that, based on the definition of an enzymatic activity unit for α-amylase (line 727-729), the enzyme used potato starch, flours extracted from the corn hybrids under study, and the purified inhibitors as substrates. Based on these results, a second experiment was performed. The results are presented in Table 2. The treatment involved incubating the enzyme with the flours and inhibitors as substrates (details are provided in section 4.13). Once the reaction was stopped, the resulting solution was used to measure its inhibitory effect on the aspartyl protease EAP1 and the xylanase SRXL1. It was observed that the flours and purified inhibitors lost their ability to inhibit the aforementioned enzymes after being degraded by α-amylase. Two controls were established: in the first, α-amylase was heat-inactivated before incubation with the flours and purified inhibitors. In the second control, distilled water was used instead of the enzyme. In both controls, the flours and purified inhibitors retained their inhibitory activity.
To prevent confusion, the microorganism's name was removed from the subtitles, and the materials and methods specified that α-amylase is a commercial enzyme produced by B. amyloliquefaciens (Lines 300, 323, 715, 721, 731).
Comment 10:
Overall, the authors have used many enzyme inhibitors abbreviations, some they have used even at first mention. I would suggest to list these abbreviations at the end or use full name and then abbreviations.
Response:
The changes were made. Lines 78, 88, and 506.
Comment 11:
Please add conclusions and future directions
Response:
The last paragraph of the discussion outlines the conclusion of the work and potential future directions (Lines 547-554).
Corrections are highlighted in yellow in the manuscript.
Your comments and suggestions will contribute significantly to improving this manuscript.
Thank you for your observations,
Dr. Yuridia Mercado-Flores
Round 3
Reviewer 3 Report
Comments and Suggestions for Authors
After clarifying the answers, everything became much clearer, thanks to the authors. The article can be printed.